# Apoptosis, a Metabolic “Head-to-Head” between Tumor and T Cells: Implications for Immunotherapy

**DOI:** 10.3390/cells13110924

**Published:** 2024-05-27

**Authors:** Ornella Franzese, Pietro Ancona, Nicoletta Bianchi, Gianluca Aguiari

**Affiliations:** 1Department of Systems Medicine, University of Rome Tor Vergata, Via Montpellier 1, 00133 Rome, Italy; franzese@uniroma2.it; 2Department of Translational Medicine, University of Ferrara, Via Fossato di Mortara 70, 44121 Ferrara, Italy; pietro.ancona@unife.it; 3Department of Neuroscience and Rehabilitation, University of Ferrara, Via F. Mortara 74, 44121 Ferrara, Italy; gianluca.aguiari@unife.it

**Keywords:** T cells, apoptosis, cancer

## Abstract

Induction of apoptosis represents a promising therapeutic approach to drive tumor cells to death. However, this poses challenges due to the intricate nature of cancer biology and the mechanisms employed by cancer cells to survive and escape immune surveillance. Furthermore, molecules released from apoptotic cells and phagocytes in the tumor microenvironment (TME) can facilitate cancer progression and immune evasion. Apoptosis is also a pivotal mechanism in modulating the strength and duration of anti-tumor T-cell responses. Combined strategies including molecular targeting of apoptosis, promoting immunogenic cell death, modulating immunosuppressive cells, and affecting energy pathways can potentially overcome resistance and enhance therapeutic outcomes. Thus, an effective approach for targeting apoptosis within the TME should delicately balance the selective induction of apoptosis in tumor cells, while safeguarding survival, metabolic changes, and functionality of T cells targeting crucial molecular pathways involved in T-cell apoptosis regulation. Enhancing the persistence and effectiveness of T cells may bolster a more resilient and enduring anti-tumor immune response, ultimately advancing therapeutic outcomes in cancer treatment. This review delves into the pivotal topics of this multifaceted issue and suggests drugs and druggable targets for possible combined therapies.

## 1. Introduction

Cancer, a severe pathology, is the second cause of worldwide death after cardiovascular diseases [1]. Currently, the main goal of fighting cancer is based on drugs able to inhibit cell proliferation and stimulate tumor-programmed cell death, including apoptosis. This kind of cell death is characterized by different morphological events that lead to DNA condensation and fragmentation as well as plasma membrane blebbing, which is involved in the formation of apoptotic bodies [2].

Apoptosis can be initiated through two primary mechanisms: the extrinsic and intrinsic pathways [3]. The extrinsic pathway is related to various cell death-inducing ligands, including FAS. The FAS–FAS ligand (FAS-L) pathway contributes to maintaining T-cell homeostasis by inducing apoptosis upon T-cell activation [4]. As described by Zhu et al. [5], various cancers, including melanoma, lung, hepatocellular, and colon carcinomas, express FAS-L, potentially inducing apoptosis in FAS-expressing immune cells. Moreover, FAS-L expression can be induced by various cytokines in activated T cells and neighboring cells, acting as an immune checkpoint (IC) mechanism to prevent immunopathology from excessive T-cell activation [6]. The activation of this extrinsic pathway enhances the activity of caspase-8 that, in turn, triggers caspase-3, leading to apoptotic cell death.

The intrinsic pathway expects the mitochondrial outer membrane permeabilization, which promotes the translocation of cytochrome c from the mitochondrial intermembrane space to the cytosolic compartment. Cytochrome c then stimulates the formation of a caspase-activating complex assembly between inactive caspase-9 and the apoptosome. This complex promotes the activation of caspase-9, which triggers the effector caspases-3, 6, and 7, leading to DNA fragmentation and apoptotic cell death. This cascade of events is regulated by the balance between anti-apoptotic and pro-apoptotic proteins, B-cell lymphoma 2 (BCL-2), and BAX, respectively [2].

In normal conditions, apoptotic cells are removed by phagocytic cells (efferocytosis), allowing the discharge of aged, damaged, and out-of-control cells, thus facilitating tissue regeneration [7]. Since cancer cells show uncontrolled proliferation, migration, and invasion, the induction of apoptosis has been proposed as a promising therapeutic strategy to kill tumor cells. Many drugs have been designed to induce apoptosis, such as inhibitors of anti-apoptotic BCL-2-like proteins, mitogen-activated protein kinase (MEK) antagonists, and mouse double minute 2 homolog (a p53 downregulator) [2]. However, the molecules released by apoptotic cells not only activate phagocytes for their removal but, in some cases, can induce pro-tumorigenic signals. In fact, apoptotic tumor cells can attract monocytes, which may differentiate into macrophages that, in turn, produce and release tumor-promoting factors causing cancer progression [8].

Published data suggest a controversial role of apoptosis in cancer; in particular, the relationship between apoptotic cells and the immune system is not fully understood. Apoptosis appears as a pivotal player in modulating the intensity and duration of anti-tumor T-cell responses while emerging as a critical contributor to T-cell dysfunction [9]. Therefore, an unrestrained use of pro-apoptotic strategies must carefully avoid triggering apoptosis in T cells from the tumor microenvironment (TME). Indeed, the susceptibility of T lymphocytes to apoptosis ultimately influences the magnitude and efficacy of the anti-tumor T-cell response. Specifically, targeting molecular pathways implicated in the regulation of T-cell apoptosis can enhance the endurance and functionality of T cells to bolster a more resilient and enduring anti-tumor immune response, leading to sustained and effective anticancer treatments. In this context, harnessing apoptosis in cancer cells while preserving or enhancing the survival and functionality of T cells within the TME, emerges as a promising strategy to enhance the response to IC blockade (ICB).

Herein, we describe the latest updates on crosstalk among apoptotic cells, the tumor environment, and immune cells, with particular attention to anticancer therapeutic strategies. Additionally, we consider the interference of cancer cell metabolism in the TME, how the metabolic renewal can drive resistance conditions to overcome apoptosis, as well as the relationship with the other interlayers in cancer control by the immune cell system and the effects of neighboring tissue cells. With this aim, we consulted the PubMed-NCBI database (https://pubmed.ncbi.nlm.nih.gov/, accessed on 20 March 2024), formulating the following query: “(((“apoptosis”) OR (“apoptotic”)) AND ((“cancer”) OR (“tumor”)) AND ((“tumor microenvironment” OR/“extracellular matrix”)) AND ((“metabolism”) OR (“metabolome”) OR (“metabolic”)) AND (“immune cell system”) [MeSH Terms]). Further research included (“T-cell apoptosis”) AND (“anti-tumor response”) AND (“T-cell homeostasis”) OR (“T-cell metabolism”))”. The investigation focused on several key aspects: fostering anti-tumor T-cell mediated responses, the influence of T-cell metabolism on susceptibility to apoptosis, and strategies for enhancing response to immunotherapies. This involved safeguarding T cells from apoptosis while bolstering the generation of long-lived memory T cells, optimizing the efficacy of IC inhibitors (ICIs) and chimeric antigen receptor (CAR) T-cell therapy.

Throughout the manuscript, we focused on the latest research findings, emphasizing their potential therapeutic applications. We consistently cited the relevant literature to support our discussions, ensuring that our analysis aligns with current advancements in the field.

## 2. Crosstalk between Apoptotic and Immune Cells Affects the Fate of Cancer

Apoptosis is involved in normal cell turnover, which induces cell death in response to a variety of stimuli. However, when dysregulated, it leads to the development of different diseases, including cancer. It is well known that apoptotic cells release a variety of molecules affecting the local microenvironment and can attract different types of cells, such as macrophages, dendritic cells (DCs), fibroblasts, and epithelial and endothelial cells [7].

Apoptotic cells produce a complex of factors that promote proliferation tissue re-modeling and repair, as well as immune signals, which suppress inflammation [7]. However, molecules released from apoptotic cells and phagocytes within the TME can promote cancer progression and immune escape. These can re-program tumor-associated macrophages (TAMs) in pro-tumorigenic cells, enhancing cell invasion and migration, thus contributing to metastasis [2,7].

TAMs constitute one of the main tumor-infiltrating immune cell types and are di-vided into two functionally different subtypes: the classical activated M1 macrophages and the alternatively activated M2 macrophages. The first subset exerts anti-tumor functions, mediating cytotoxicity and antibody-dependent cell-mediated cytotoxicity (ADCC) to induce tumor cell death. The latter has an opposite function and can stimulate tumor recurrence and metastasis by inhibiting T-cell-mediated anti-tumor immune response [10].

### 2.1. Molecular Signals Generated by Apoptotic Cells in Cancer

It is well known that phosphatidylserine (PS) translocates to the cell surface of apoptotic cells and interacts with specific receptors such as brain-specific angiogenesis inhibitor 1 (BAI1), T-cell/transmembrane immunoglobulin and mucin domain containing (TIM)-4 and Stabilin-2 or other receptors including Mer tyrosine Kinase (MerTK) localized on the plasma membrane of phagocytes. These interactions activate signaling pathways leading to the removal of dying cells [11]. More recently, it was reported that PS induces polarization and accumulation of M2-like macrophages that contribute to the immunosuppressive TME. PS acts by activating TIM-4 receptors, focal adhesion kinase-SRC-STAT3 signaling, and enhancing the expression of the histone demethylase Jumanji domain-containing protein 3 in tumor models [12]. Moreover, many other molecules are released by apoptotic cells in the microenvironment, including immunogenic proteins such as histones, which activate Toll-like receptors inducing cytokine production [13]. In addition, the induced apoptosis causes the release of neutrophil-attracting chemokines such as interleukin-8 (IL-8) in colorectal cancer cells, which, via macrophage interaction, contribute to promoting an immunologically unfavorable TME (Figure 1) [14].

Another chemotactic factor released by apoptotic cells is the S1P that potently stimulates the chemotaxis of monocytic THP-1 and U937 cells, as well as primary monocytes and macrophages. S1P serves as a “find-me” signal to attract phagocytes and engulf apoptotic cells in order to prevent necrosis and inflammation [15]. “Find-me” signals may be activated by CRT produced by cancer apoptotic cells. In particular, cell surface-anchored CRT interacts and collaborates with other molecules in order to eliminate apoptotic dead cells and mediate ICD. In addition, exogenous CRT, released by apoptotic cells is internalized by THP1 macrophages, triggering a pro-inflammatory response [7,16].

Surface exposure of CRT combined with heat shock protein (HSP) 70 and HSP90 release was also observed in primary human acute myeloid leukemia cells cultured in the presence of cytotoxic drugs and all-trans retinoic acid [17]. Thus, apoptotic cells produce damage-associated molecular patterns, including surface-exposed CRT and ATP, secreted after chemotherapy agent exposition, which, in turn, appears to modulate ICD [18]. The release of ATP from dying cells also constitutes a “find-me” signal for the recruitment of DCs, inducing a pro-inflammatory response. Importantly, ATP release from apoptotic cells occurs through a caspase and pannexin-1-dependent lysosomal exocytosis mechanism [19].

ICD has been defined as a cell death modality that generates a protective immune response against dead-cell antigens and is able to remove demised cells after chemotherapeutic drug administration [19]. ICD is mainly triggered by three well-known events such as CRT exposure, ATP secretion, and the release of the HMGB1 (Figure 1). The absence of one of these ICD hallmarks strongly reduces the efficacy of chemotherapeutic response [19].

In light of these observations, therapeutic treatments should be addressed to induce apoptotic cell death involving the activation of ICD. In this regard, it was reported that drug activation of receptor-interacting serine/threonine-protein kinase 1 (RIPK1) can trigger cell death that evokes the immune system response against cancer. In particular, RIPK1-mediated cell death significantly increases the activation of CD8^+^ T and NK cells as well as it potentiates the effect of ICD in soft-tissue sarcoma in vivo models [20]. Moreover, the combined treatment with radiotherapy and ataxia telangiectasia and Rad3-related kinase inhibitors enhances the release of HMGB1 and ATP, thus contributing to the activation of anti-tumor immunity in different cancer cell types [21]. In addition, the combination of doxorubicin and the STAT3 inhibitor stattic is able to increase ICD, leading to DCs’ functional maturation and IL-12 secretion in melanoma and colon cancer cells, suggesting that this strategy might trigger a cell-mediated immune response [22].

Furthermore, treatment with the proteasome inhibitor bortezomib induces ICD, leading to membrane surface CRT exposition in dying multiple myeloma (MM) cells and causing phagocytosis of tumor cells by DCs. Bortezomib stimulates MM cell immunogenicity in a mechanism involving the activation of the cGAS/STING pathway and type I IFNs (interferons) production. Moreover, STING agonists strongly potentiate bortezomib-induced ICD; therefore, the combined treatment with bortezomib and STING agonists could increase tumor-specific immunity and improve the outcome of patients with MM [23]. Consistently, treatment with carfilzomib, another proteasome inhibitor, is able to activate ICD in myeloma cells, improving the prognosis of MM patients by cytotoxic effects as well as through the triggering of immune memory response, especially in combination with other therapies [24].

Finally, it was reported that the administration of selenium nanoparticles in colon cancer cells induces apoptosis and activates ICD by the translocation of CRT and ERp57, the release of HMGB1 and ATP, and the secretion of pro-inflammatory cytokines. This treatment seems to be an efficient strategy to kill tumor cells by inducing apoptotic cell death and eliciting immune responses [25].

Taken together, these observations suggest that apoptotic cancer cells play an active role in inducing anti-tumor immunity, but chemotherapy-induced apoptosis cannot always trigger an immune response.

In fact, treatment with tamoxifen, paclitaxel, and other drugs may promote therapy resistance by M2-like macrophage activation in different cancer types. Therefore, strategies addressed to target M2-like macrophages are crucial to overcome anti-tumor drug resistance [26].

To enhance clarity, all these concepts have been schematized in Figure 1, including relevant drugs discussed in the subsequent section.

### 2.2. Mechanisms of Apoptosis Escape Devised by Cancer Cells

Despite the induction of apoptosis is one of the most promising goals for the treatment of cancer, the innate and acquired resistance to anticancer drugs, including chemotherapy-induced apoptosis, is a major problem in cancer treatment and disease control [20,27]. Cancer cells have adopted different strategies to evade chemotherapy-induced cell death, including genetic and epigenetic mechanisms leading to the increase in anti-apoptotic protein expression, the impairment of pro-apoptotic signals, and mutations in apoptotic-related genes [2].

Mechanisms to elude apoptosis devised by cancer cells include altered expression of FAS and its ligand FAS-L, which can activate apoptosis signaling and induce cell death. In fact, tumor cells can escape FAS-mediated apoptosis by downregulating the FAS/FAS-L signaling pathways, which seem to be associated with poor prognosis in breast cancer patients [28]. Recent studies have reported that the FAS-associated death domain (FADD) is critical in the regulation of cancer cell proliferation, and the suppression of FADD expression is a mechanism adopted by cancer cells to escape from apoptosis. Consistently, the intracellular delivery of FADD protein in cancer cells recovers apoptotic signaling, increasing cell death. Therefore, the delivery of FADD proteins into cancer cells could represent a promising therapeutic approach for tumor therapy [29].

Moreover, the dysregulation of other apoptosis-associated proteins such as BCL-2, survivin and caspases may affect apoptosis in breast cancer as well as in other neoplasms. BCL-2 overexpression in breast cancer cells prevents apoptosis and is associated with neoplastic transformation and enhanced cellular survival. Decreased expression and/or activation of caspases may represent another strategy used by tumor cells to inhibit apoptosis in cancer [28,30]. Interestingly, cancer cells may acquire resistance to treatment with BCL-2 inhibitors by mutational mechanisms and through the hyper-phosphorylation of BCL-2 family proteins. In fact, resistance phenomena after treatment with the BCL-2 inhibitor venetoclax in lymphoid malignancies have been observed, mainly due to the hyper-phosphorylation of MCL-1, BCL-2, BAD, and BAX proteins [31]. Nevertheless, treatment with venetoclax is currently used for hematological malignancies, and it is undergoing clinical trials in hematological and solid tumors. Moreover, the use of next-generation inhibitors of anti-apoptotic proteins BCL-2, BCL-XL, and MCL-1 induces apoptosis in in vitro and in vivo tumor models [31]. The overexpression of BCL-XL, a member of the BCL-2 family showing anti-apoptotic properties, confers an oncogenic dependency in malignant pleural mesothelioma (MPM). Interestingly, targeting BCL-XL combined with autophagy inhibitors increases tumor cell apoptosis, suggesting a potential therapeutic strategy for this cancer [32].

Another study reports that mitochondrial apoptosis resistance is strongly associated with activating mutations of JAK3 in T-cell acute lymphoblastic leukemia (T-ALL). JAK3-activating mutations inhibit apoptosis triggering the MEK and ERK pathway that ultimately leads to the phosphorylation of BCL-2. Treatment with the JAK3 inhibitor tofacitinib in combination with conventional chemotherapeutics is significantly more effective than monotherapy in a JAK3 mutant T-ALL mouse model [33]. Apoptosis resistance may also be associated with mutations in the *PIK3CA* gene that confer resistance to chemotherapy in triple-negative breast cancer (TNBC) by inhibiting apoptosis and activating the PI3K/AKT/mTOR signaling pathway. Importantly, the prognostic outcome of TNBC patients carrying PIK3CA mutations indicates that these subjects are more susceptible to relapse and metastasis. Therefore, the molecular targeting of PIK3CA mutants or the related pathway could be a strategy for the treatment of PIK3CA-mutated TNBC patients. However, clinical trials using inhibitors of PI3K and AKT in metastatic breast cancer with PI3CA mutations showed unsatisfactory results; thus, new inhibitors should be tested in the future [34].

Mutations in the tumor suppressor gene *TP53*, defined as the guardian of the genome, are among the major mechanisms devised by tumor cells to escape from apoptosis. In fact, *TP53* is found to be mutated in about 50% of human malignancies, and its function is almost abrogated in the rest of the cancers. Since gain of function (GOF) p53 mutants are strongly associated with tumor progression and drug resistance, they represent possible targets for developing novel cancer therapies [35]. Currently, different compounds capable of p53 reactivation or the destabilization of mutant p53 are being investigated. Several of them, such as APR-246, COTI-2, SAHA, and PEITC, are already approved for clinical trials [36]. These compounds can inhibit or cause the degradation of p53 mutants, but data regarding clinical trials are still incomplete. Nevertheless, treatment with p53-mutant inhibitors combined with other drugs seems to be promising; in fact, combined treatment with eprenetapopt and azacytidine (clinical trial NCT03072043) improves the clinical outcome in patients with myelodysplastic syndromes and oligoblastic acute myeloid leukemia [36]. Another approach to preserve wild-type p53 is the inhibition of proteins that lead to its degradation, such as MDM2 and MDMX. Several clinical trials using drugs targeting these molecules have been concluded with promising results; however, no MDM2 or MDMX inhibitor has been approved by the FDA [37].

Interestingly, p53 dysfunction also activates inflammation and supports tumor immune evasion, promoting cancer progression. Therefore, treatment with agents that prevent wild-type p53 degradation could inhibit immunosuppression and enhance anti-tumor immunity. The pharmacological treatment of lymphoma and melanoma mouse models with the p53 activator nutlin-3a induces anti-tumor immunity and tumor regression. The reactivation of p53 reverts the immunosuppression in the TME and activates ICD [38]. Other approaches to restore p53 function or to inhibit p53 GOF in the TME involve the use of specific antibodies that target p53 mutants and the reactivation of wild-type p53 by gene therapy and by ICIs. However, further investigations to verify the efficacy of these approaches must be carried out [39].

Epigenetic factors may strongly affect apoptosis in cancer cells; the major epigenetic alterations are mainly caused by aberrant DNA methylation, histone modification, chromatin remodeling, and microRNA (miRNA) expression. In different cancer types, the hypermethylation of tumor suppressor and pro-apoptotic gene promoters was observed. DNA promoter methylation leads to the downregulation of several pro-apoptotic factors, such as FAS, Caspase 8 and 10, BAX, BAD, PUMA, and other pro-apoptotic genes in different solid and blood tumors [40]. The hypermethylation in other key genes involved in carcinogenesis was reported; in fact, the silencing of *BRCA1* gene expression by DNA methylation is associated with advanced breast cancer [41]. Moreover, the aberrant methylation of *RARβ2*, *DAPK*, *hMLH1*, and p14 genes is associated with breast cancer susceptibility [42].

Given the importance of DNA methylation dysfunction in cancer, many drugs that target proteins involved in this process have been developed. Azacitidine, decitabine, guadecitabine, and 4-thio-2-deoxycytidine are DNA (cytosine-5)-methyltransferase 1 inhibitors designed to reduce DNA methylation. These drugs were approved in clinical trials for the treatment of advanced solid tumors, myelodysplastic syndromes, and acute myeloid leukemia [40].

Chromatin remodeling plays important roles in normal physiology and diseases, particularly cancer. In this regard, it was observed that oncogenic chromatin remodeling and YAP-dependent transcription induced by the mTOR Complex 1 (mTORC1)– AT-rich interactive domain-containing protein 1A (ARID1A) axis promotes cancer cell growth and tumor development in different models for hepatocellular carcinoma (HCC). YAP-targeted therapies could represent a new option for the treatment of HCC [43]. Moreover, the chromodomain-helicase-DNA-binding protein 4 (CHD4), acting as a chromatin remodeler, seems to be associated with platinum therapy resistance in ovarian cancer. Interestingly, treatment with the CHD4/SMARCA5 inhibitor ED2-AD101 showed synergistic interactions with cisplatin therapy. Therefore, CHD4 inhibition could be a novel therapeutic strategy in combination with platinum agents [44].

Altered histone modifications are related to tumorigenesis; in fact, reduced acetylation of H3 and H4 due to high histone deacetylase (HDAC) activity may silence tumor suppressor genes and stimulate cancer progression as well as apoptosis resistance. The histone methyltransferase NSD2 is able to regulate tumorigenesis and chemosensitivity in osteosarcoma. In particular, NSD2 inhibits apoptosis, modulating the expression of the apoptotic proteins BCL-2 and SOX2 through the ERK and AKT pathways. These findings suggest that NSD2 might be a new target for combined chemotherapy in osteosarcoma [45]. Moreover, it was reported that HDAC2 and enhancer of zeste homologue 2 (EZH2) proteins induce epigenetic alterations leading to pyruvate dehydrogenase kinase 1 (PDK1) upregulation through miR-148a silencing. PDK1 overexpression causes tumor development and adriamycin resistance in breast cancer, and thus, the HDAC2/EZH2/miR-148a/PDK1 axis may represent a potentially promising therapeutic target [46].

Finally, the aberrant expression of miRNAs plays a key role in the initiation and progression of cancer. MiRNAs are also regulated by epigenetic factors such as DNA methylation and histone modifications. In cancer cells, it was found that miR-17-92 overexpression promotes cell proliferation, while the miR-17-20 cluster, which represses cyclin D1 expression and suppresses breast cancer cell proliferation, is downregulated in breast tumors. Moreover, miRNAs are also associated with the regulation of apoptosis in cancer. Indeed, many identified miRNAs with both anti-apoptotic and pro-apoptotic properties regulate the expression of apoptotic genes such as phosphatase and tensin homolog (PTEN), caspase-9 and BCL-2, resulting in apoptosis resistance and tumor expansion [47].

As altered miRNA expression is involved in carcinogenesis, the delivery of miRNA or anti-miRNA sequences in tumor cells is considered an attractive option for cancer treatment. Currently, clinical trials based on miRNA therapy are still in the primary stage and adverse effects should be evaluated. Nevertheless, MRX34 is the first miRNA-based therapy applied in patients with primary liver cancer. It acts by mimicking miR-34a, which shows anti-tumor properties acting downstream on the *TP53* gene [48].

Another way to inhibit apoptosis adopted by cancer cells is the expression of inhibitor of apoptosis (IAP) proteins, altered in several cancer types and implicated in chemotherapy resistance. In fact, the overexpression of IAP proteins such as cIAP-2, survivin, and XIAP in pancreatic cancer was reported [49]. In addition, survivin was found to be selectively overexpressed in various tumors, including head and neck squamous cell carcinoma (HNSCC). Therefore, targeting this molecule might be considered a good therapeutic option for cancer treatment [50].

An additional mechanism involved in therapy resistance is represented by signals released by chemotherapy-induced apoptosis, which may promote cell re-population by inducing a compensatory proliferation mechanism called AiP. This can be induced by the translocation of PS to the plasma membrane and the release of S1P by apoptotic cells, leading to the production of prostaglandin G E2 type (PGE2), vascular endothelial growth factor (VEGF), epidermal growth factor (EGF), transforming growth factor-β (TGF-β), matrix metalloproteinase 9, and anti-inflammatory cytokines (Figure 1). These tumor-promoting factors inhibit anti-tumor immunity, promote angiogenesis, and stimulate cancer progression. Moreover, some chemotherapeutic drugs currently approved for cancer treatment, such as cisplatin or 5-fluorouracil, are unable to activate ICD and might activate drug-resistance mechanisms [51]. Conversely, treatments that induce the release of CRT, ATP, and HMGB1 lead to the activation of ICD, causing cell demise and improving the anticancer response (Figure 1) [7,8].

As described above, survivin is an apoptosis inhibitor peptide belonging to the IAP family of proteins. The expression of this peptide is linked to poor prognosis in glioblastoma (GBM) as well as in other cancer types. Consistently, the combined treatment with temozolomide and the peptide-vaccine conjugate SurVaxM that may activate an immune response against survivin seemed to give apparent clinical benefits in GBM patients enrolled for a Phase IIa clinical trial [52]. Moreover, another suvivin vaccine employed in preclinical studies using a mouse model for triple-negative breast cancer was tested. The immunization of mice with adjuvanted survivin peptide microparticles decreased the growth rate of primary tumors compared with control mice, suggesting that T-cell immunotherapy targeting survivin could be an applicable approach for the treatment of triple-negative breast cancer [53].

As previously emphasized, the induction of apoptosis after chemotherapy may generate an anti-inflammatory response by outer plasma membrane PS exposition and MerTK oncogene activation. So, the inhibition of both PS signaling and MerTK activation could be an additional strategy for cancer treatment (Figure 1). Interestingly, the combined therapy between a TLR9 activator (CpG-2722) and a PS-targeting drug (BPRDP056) exerts anti-tumor functions enhancing immune response in an orthotopic head and neck cancer animal model [54]. Moreover, treatment with anti-PS antibodies enhances the anti-tumor efficacy of radiation therapy and improves overall survival, increasing pro-inflammatory tumor-associated macrophages in a preclinical melanoma model [55]. Nevertheless, the combined treatment with bavituximab, an immunomodulator that targets PS, and sorafenib, although well tolerated, did not demonstrate evidence of improved efficacy compared to patients treated with sorafenib alone in a single-arm Phase II trial of advanced hepatocellular carcinoma (HCC) [56]. Given the encouraging findings obtained in tumor preclinical models, further clinical trials using other PS inhibitors should be performed.

The inhibition of MerTK induces the accumulation of cancer apoptotic cells and triggers a type I interferon response. Moreover, treatment with an anti-MerTK antibody stimulates T-cell activation and synergizes with ICIs, including monoclonal antibodies (mAbs) targeting programmed death protein 1 (PD-1) or PD-1 ligand (PD-L1) in the MC38 colon carcinoma mouse model. The inhibition of MerTK increases tumor immunogenicity and potentiates anti-tumor immunity, enhancing cancer immunotherapy [57]. Currently, the drug sitravatinib, an inhibitor of tyrosine kinase (TK) that targets different TK, including MerTK, has been used in different clinical trials. Results obtained from treating patients with sitravatinib and nivolumab in a Phase III study of advanced non-squamous non-small-cell lung cancer (NSCLC) as well as in a Phase II trial for advanced renal carcinoma did not statistically increase the objective response rate [58,59].

Interestingly, the loss of function of MerTK increases M1-like anti-tumor phenotypes and reduces M2-type pro-tumor macrophages in the high-MYC prostate cancer mouse model as well as decreases M2-mediated efferocytosis in prostate cancer LNCaP cells [60]. In this regard, it is known that M2 TAMs may affect the outcome of tumors by reducing the immune response and promoting cancer progression. Therefore, the removal of these cells could improve the efficacy of chemo/immunotherapy. Currently, treatment with nanoliposome C6 ceramide reduces the number of TAMs and their ability to inhibit the anti-tumor immune response in mouse models for hepatocellular cancer. Thus, this molecule could increase the efficacy of immune therapy in patients with hepatocellular carcinoma [61]. As already mentioned, the reprogramming of M2 TAMs into M1 phenotypes could be a successful option for cancer therapy. In fact, treatment of M2-type macrophages with a combination of the DNA methylation inhibitor 5-aza-dC and the histone deacetylation inhibitor trichostatin A decreases the levels of M2 macrophages, increasing the M1 subtype, and sensitizes the tumor cells to paclitaxel. Moreover, the combined treatment inhibits tumor growth and improves anti-tumor immunity in the TME [62].

However, chemotherapy-induced apoptosis may activate immunosuppressive TAMs, which secrete different soluble factors that interact with cancer stem cells (CSCs), saving them from environmental stress and immune response [63]. Moreover, it was reported that glioblastoma CSCs produce immunosuppressive cytokines, including TGF-β, which stimulates the functional polarization of pro-tumorigenic TAMs. Other immunosuppressive cytokines secreted by CSCs include IL-4 and IL-13 that promote M2-like macrophage maturation [63]. In addition, M2 TAMs can enhance CSC characteristics in HCC cells, resulting in enhanced resistance against sorafenib [64]. Interestingly, it was found that the inhibition of STAT3 and NF-κB in macrophages abolished the TAM-promoted stemness in several cancer types [63]. Consistently, treatment with M1 macrophage extracellular vesicles containing drugs such as oxaliplatin and retinoic acid reduced *STAT3*, *NF-κB*, and *AKT* gene expression as well as the levels of IL-10, TGF-β, and CCL2. shifting M2-like macrophages to M1 phenotypes. This treatment enhanced apoptosis in different colorectal cancer cell lines and reduced tumor growth and metastasis in mice of allograft and peritoneal colorectal cancer models [65].

Finally, it was reported that *STAT3* knockdown promoted sorafenib-induced ER stress-induced apoptosis in HCC cells enhancing the anti-tumor function of CD8^+^ T and NK cells. The combined treatment with sorafenib and *STAT3* knockdown may affect the TME via the cGAS-STING-type I IFNs axis of DCs, activating anti-tumor immune responses [23]. Furthermore, treatment with domatinostat, a small-molecule inhibitor targeting HDAC class I, induces G2/M arrest and apoptosis in Merkel cell carcinoma (MCC) cells. Domatinostat exerts direct anti-tumoral effects and restores HLA class I surface expression on MCC cells, restoring the killing properties of cognate cytotoxic T cells [66]. The observations reported here indicate that chemotherapeutic strategies addressed to induce apoptosis should be carefully evaluated in order to stimulate ICD and minimize the release of factors that promote apoptosis-induced proliferation and/or immune response suppression, as well as CSCs activation, which may promote tumor progression and immune escape resulting in therapy resistance.

## 3. A More In-Depth Insight into the Role of Apoptosis in the Immune Landscape of the TME

### 3.1. Exploring Key Subsets in the TME

Tumor-infiltrating T-cell apoptosis plays a crucial role in the complex dynamics of the TME, with significant implications for treatment resistance in cancer. As T cells infiltrate the tumor, they encounter an unfriendly and immunosuppressive milieu that can trigger apoptotic pathways, leading to their demise. Understanding the mechanisms underlying this phenomenon is essential for devising effective therapeutic strategies. The critical interactions between immune cells and tumors involve competition for vital nutrients like glucose and amino acids, which has far-reaching implications, compromising the functional state of immune cells while impacting disease progression and responses to immunotherapies, including ICB.

Understanding the immune landscape in tumors is crucial for comprehending the intricate interplay between immune responses and metabolic shifts in the TME. However, numerous studies have already detailed the diverse immune components within the TME, which exceeds this review’s scope.

Cancer progression relies on a complex relationship among various cells in the TME, including leukocytes, fibroblasts, stromal cells, and vascular endothelial cells [67]. Maintaining a balance between pro-inflammatory and anti-inflammatory responses is crucial. While Th1 CD4^+^ cytokines support CD8^+^ cytotoxic T lymphocytes’ tumoricidal activity, the Th2 response, characterized by the release of TGF-β, IL-4, IL-13, and IL-10, hampers this function. As described above, circulating monocytes can differentiate into TAMs, exhibiting either pro-inflammatory M1 or anti-inflammatory M2 features shaped by TME cues [68].

Myeloid-derived suppressor cells (MDSCs) are pivotal in creating an immunosuppressive TME, aiding tumor progression. A heterogeneous population of immature myeloid cells, MDSCs exert their impact by fostering immune tolerance, suppressing the activation of NK and T cells while promoting the expansion of regulatory T cells (Tregs) and tumor metastasis by inducing an epithelial–mesenchymal transition (EMT) [69].

Tumor-infiltrating lymphocytes (TILs), particularly CD8^+^ T cells, are critical for recognizing and eliminating cancer cells, serving as a crucial predictor of cancer prognosis. Seminal studies focused on evaluating generic T-cell infiltration, as reported by Camus et al. [70], have classified cancer lesions as either hot (with significant immune cell infiltration and inflammation) or cold (displaying poor infiltration and minimal inflammation), offering valuable insights into the immune dynamics within the TME and their clinical implications.

A large subset of tumor-infiltrating CD8^+^ T lymphocytes are tissue-resident memory (T_RM_) cells, crucial for robust anti-tumor immunity [71]. Found in different human cancers, including NSCLC, T_RM_ cells are characterized by the expression of CD103 and CD69, which contribute to their residency features [72,73].

Additional cellular clusters known as tertiary lymphoid structures, which consist of an organized arrangement of a follicular area containing CD20^+^ B cells encircled by a CD3^+^ T-cell zone, facilitate the on-site activation of anti-tumor lymphocytes. These cellular organizations support antigen presentation by DCs while fostering the production of effector and plasma cells [74].

Tumor-infiltrating T cells can become exhausted, a condition characterized by low effector function and persistently high levels of IC receptors [75,76], including PD-1, cytotoxic T lymphocyte antigen 4 (CTLA-4), TIM-3, lymphocyte activation gene 3 (LAG-3), T-cell immune-receptor with Ig and ITIM domains (TIGIT), along with others [77]. Exhaustion involves a dynamic evolution from a “progenitor” to a “terminally exhausted” state rather than an ultimate T-cell deactivation [78], with distinctive patterns of IC co-expression [79].

Tregs play a critical function within the TME, sustaining immune homeostasis and mitigating excessive inflammation by inhibiting immune responses directed towards self-antigens [80]. This regulatory function extends also to situations such as autoimmunity and allograft rejection [81].

Tregs contribute to T-cell exhaustion, inhibiting anti-tumor responses through various mechanisms, including APC inhibition, cytokine depletion, and production of immunosuppressive factors, ultimately promoting tumor proliferation, and reducing responses to immunotherapies [82]. Although Forkhead box protein P3 (FOXP3) serves as a crucial regulator in both the development and suppressive function of CD4^+^ Tregs, a complicated categorization involves subtypes such as CD4^+^CD25^+^FOXP3^+^, CD4^+^CD25^−^FOXP3low, CD4^+^CD25highCD125low, CD4^+^CD25highCD45ROhigh, and CD4^+^CD25^+^CD62LlowCD44high [80]. Unlike the well-established biology of CD4^+^ Tregs, the heterogeneous features of CD8^+^ Tregs are still being revealed and involve the TGF-β-dependent expression of CD103 and FOXP3 in aggressive tumors [83]. Furthermore, an age-dependent accumulation of CD8^+^CD28^−^ T cells exhibiting suppressive activity has also been documented [84].

In conclusion, the TME complex dynamics are intricately shaped by diverse immune cells, each fulfilling distinct roles in balancing pro-tumorigenic and anti-tumorigenic responses. A comprehensive understanding of their interplay with tumor cells is crucial for devising targeted therapeutic strategies to harness tumor-specific immune responses.

### 3.2. Role of Apoptosis in the Homeostatic Control of Anti-Tumor T-Cell-Mediated Responses

Apoptosis emerges as a pivotal player in modulating the intensity and duration of anti-tumor T-cell responses and a critical contributor to T-cell dysfunction [9]. Nevertheless, this fine-tuned regulatory mechanism maintains the immune system balance, preventing issues like autoimmunity or immunodeficiency [85].

By impairing the survival and persistence of tumor-targeting T cells, apoptosis can play a pivotal role in shaping the clinical outcomes of immunotherapies, ultimately impacting treatment efficacy and patient responses. Indeed, recent investigations have highlighted apoptosis as a contributing feature to reduced T-cell infiltration in tumors, alongside factors like poor tumor antigenicity. Zhu et al. suggest that macrophages’ rapid clearance of apoptotic bodies may obscure apoptosis detection in vivo or in patient samples, potentially masking previous T-cell presence in “cold” tumors [5].

While the extrinsic and intrinsic apoptotic pathways can operate independently, their extensive crosstalk amplifies the death signal [9]. In T cells, the extrinsic pathway is initiated by death-inducing cytokines, such as TNF-α, the CD95 ligand, or the TNF-related apoptosis-inducing ligand (TRAIL), binding to their respective receptors on apoptosis-sensitive cells. Pro-apoptotic signals encompass death receptor signaling and the activation of pro-apoptotic proteins. Six human death receptors, including FAS, TRAIL-R1, TRAIL-R2, TNFR1, DR3, and DR6, have been identified, as extensively described by Zhu et al. [5] and others. The FAS–FAS-L pathway maintains T-cell homeostasis by inducing apoptosis upon T-cell activation [4] but also impacts lymphocyte differentiation, especially in the memory compartment, compared to naïve CD8^+^ T cells, which instead exhibit better intra-tumor persistence [86].

On the other hand, T-cell apoptosis is controlled by a balance of pro-survival (e.g., BCL-2 and MCL-1) and pro-apoptotic (e.g., BIM, NOXA, and PUMA) factors [87], while BH3 proteins like BIM and NOXA rely on BAX/BAK for apoptosis induction. When BH3 proteins dominate, BAX and BAK permeabilize the mitochondrial outer membrane, releasing cytochrome c and activating effector caspases. BIM and NOXA mediate the peripheral deletion and accumulation of activated T cells [88], while PUMA induces antigen-specific T-cell death [89]. These proteins collaborate in regulating T-cell survival. Indeed, a simultaneous loss of PUMA and BIM protects T-cell blasts from IL-2 deprivation-induced death more effectively than the loss of BIM alone [88,90], potentially leading to severe autoimmunity and organ damage. Moreover, BIM and FAS collaborate to regulate T-cell responses, highlighting the crucial role of BH3 proteins, especially BIM, in limiting T-cell survival, as highlighted by Zhang et al. [91].

Naïve long-lived and resting T cells rely on signals such as T-cell receptor/major histocompatibility complex (TCR)/MHC and IL-7 for their homeostasis, regulated by the BCL-2 family [92]. Upon antigen encounter, naïve CD8^+^ T cells transition into effectors, leading to a dynamic immune response characterized by expansion, contraction, and memory phases. During the peak of an adaptive immune response, T cells are susceptible to restimulation-induced cell death (RICD) [93], sustained by the FAS/FAS-L pathway [4]. Effector CD8^+^ T-cell effector cells also undergo apoptosis during contraction, aided by mechanisms like cytokine withdrawal-induced death (CWID) when IL-2 levels decline (Figure 2) [94,95]. Subsequently, a subset of effector CD8^+^ T cells persists, evolving into memory cells, while a significant fraction, lacking essential cytokines, succumbs to apoptosis. IL-7 sustains memory T-cell survival by inhibiting apoptosis of IL-2-depleted activated T cells by inducing anti-apoptotic BCL-2 via JAK/STAT signaling while promoting glucose uptake and proliferation through the PI3K/AKT pathway [92,96]. Moreover, IL-15 reduces pro-apoptotic BAX and increases anti-apoptotic BCL-XL in CD8^+^ T cells, contributing to the generation of long-lived antigen-responsive cells (Figure 2) [97].

A condition known as X-linked lymphoproliferative disease (XLP-1) exemplifies how a deficiency in RICD disrupts immunological homeostasis, causing unrestrained T-cell proliferation and severe immunopathology [81]. In XLP-1 patients, a defect in the small adaptor protein (SAP) impairs the signaling lymphocytic activation molecule (SLAM), hindering the activation of pro-apoptotic molecules like FAS-L and BIM, altering T-cell RICD sensitivity [98].

Tregs may also exhibit heightened levels of both FAS-L and FAS. While FAS is expressed in Tregs from healthy individuals and cancer patients, elevated FAS-L is seen only in Tregs from cancer patients, linked to poor CD8^+^ T-cell infiltration and FOXP3^+^ Treg predominance [99]. Although FAS expression exposes Tregs to FAS-L-mediated apoptosis [100], recent research suggests they can counteract apoptotic cell death by expressing c-FLIP [101].

Various cell types in the TME, including TAMs and cancer-associated fibroblasts (CAFs), express FAS-L, contributing to intra-tumor immune dysregulation [102]. Moreover, FAS-L expression by tumor endothelium hinders CD8^+^ T-cell infiltration and fosters FOXP3^+^ Treg predominance [103], while IFN-γ-driven FAS-L expression in MDSCs enhances their accumulation and induces apoptosis in FAS-expressing T cells [104].

The pro-apoptotic effects of TRAIL are crucial, impacting different immune cells, including MDSCs and Tregs [105]. Intriguingly, CD8^+^ T cells seem unresponsive to TRAIL-induced apoptosis but susceptible to TRAIL-mediated functional inhibition [106,107].

TNFR1 and TNFR2 can act as soluble decoys inhibiting TNF-α bioactivity in the TME. TNF-α plays a dual role in regulating anti-tumor immune responses. While crucial for intra-tumor CD8^+^ T-cell cytotoxicity [79,108], it also promotes MDSC accumulation and selectively activates Tregs [105]. Thus, the outcome of using TNF-α antagonists to alleviate immunosuppression in the TME remains uncertain due to the complexity of the TNF-α roles.

Hypoxia, a key feature of the TME, plays a crucial role in tumor immune evasion, impacting T-cell survival and promoting Treg development and recruitment to the TME via the VEGFR2/VEGF axis [80]. Tregs, in turn, promote effector T-cell apoptosis by means of granzyme B and hinder their proliferation by consuming IL-2 [109].

The adenosine pathway is also crucial in the immunosuppressive network within tumors [110]. Adenosine generation occurs mainly through the successive activities of CD39 and CD73 ecto-5′-nucleotidases [111] and impairs T-cell function through the A2AR and A2BR receptors via modulation of the (cAMP)/PKA cascade [112]. Adenosine also suppresses CD8^+^ T-cell metabolism, impairing mTORC1 [113], mainly in T cells with a central memory (CM) phenotype, reducing IL-2 production, inducing CD28 loss and apoptosis through both caspase-3-dependent and -independent pathways [114].

TGF-β plays a dual role in cancer, acting both as a tumor promoter and suppressor. Indeed, while promoting the FOXP3-dependent differentiation of Tregs [115,116] and the generation of M2 macrophages, hindering overall T-cell function and inducing apoptosis in effector T cells [117], TGF-β also promotes the generation of CD8^+^ T_RM_ cells within tumors [73]. In murine models of anti-microbial immunity, TGF-β downregulates BCL-2 levels, increasing susceptibility to apoptotic signals and upregulating the pro-apoptotic protein BIM [118,119]. Moreover, TGF-β-mediated inhibition of IL-7R weakens IL-7-mediated anti-apoptotic signals, rendering T cells more susceptible to apoptosis [120].

A fine-tuned balance between co-stimulatory and inhibitory pathways regulates T-cell apoptosis to maintain immune equilibrium. Co-stimulatory molecules, including CD28, ICOS, OX40, 4-1BB, and CD27, are crucial for T-cell activation, proliferation, and survival [121]. Many anti-apoptotic molecules have been reported as crucial effectors downstream of co-stimulatory signals.

CD28 lowers the activation threshold for TCR engagement, promoting mature immunological synapse formation [122]. However, while enhancing T-cell activation and survival, CD28 also contributes to both positive and negative regulation of T cells, with multifaceted roles in protecting against intra-tumor RICD. Seminal studies have demonstrated that CD28 co-stimulation promotes IL-2 production and increases BCL-XL expression (Figure 2), enhancing the survival of activated T cells [123]. Furthermore, CD28 protects activated T cells with defective SAP from RICD by stimulating lymphocyte function-associated antigen 1 (LFA-1), which is also crucial for functional synapse formation [124,125], a mechanism that appears to be absent in CD28^−^ counterparts. These cells, lacking the LFA-1-mediated protection, are identified as a terminally differentiated effector population predisposed for clearance by apoptosis [126].

Conversely, CD28 signals can exacerbate apoptosis under conditions of excessive antigen-mediated activation, particularly when the TCR signal surpasses the activation threshold, highlighting its dual role in CD8^+^ T-cell regulation [127].

4-1BB (TNFRSF9, CD137) is a co-stimulatory molecule of the TNFR superfamily activated upon T-cell stimulation [121,128]. Interaction with 4-1BBL (TNFSF9, CD137L) on APCs triggers the recruitment of TRAF family members, generating the 4-1BB signalosome and activating NF-κB, MAPK, and ERK pathways.

Stimulation of 4-1BB also increases the expression of anti-apoptotic proteins BCL-XL (Figure 2) and BFL-1, thereby preventing RICD [129] while elevating IL-2 and IFN-γ in CD8^+^ cells and IL-2 and IL-4 in CD4^+^ cells.

CD27, a member of the TNFR family (TNFRSF7), is consistently found on CD4^+^ and CD8^+^ T cells and a specific subset of NK cells. Unlike other TNFR family members, CD27 forms a disulfide-linked homodimer at the cell surface, increasing upon T-cell activation. CD27 is stimulated by interaction with CD70 (TNFSF7, CD27L) on mature DCs, activated B and T lymphocytes, and certain hematologic malignancies, and reduces FAS-L-mediated T-cell apoptosis (Figure 2) [130] by inducing BCL-XL and PIM-1 [131], thus limiting mitochondrial dysfunction [132].

Conversely, upregulated ICs contribute to T-cell apoptosis in the TME. PD-1, identified in the early 1990s [133], impairs T-cell survival by inhibiting the activation of PI3K, crucial for upregulating BCL-XL. Interestingly, CTLA-4 ligation has minimal effects on CD28-mediated BCL-XL upregulation, emphasizing the distinct inhibitory mechanisms employed by PD-1 and CTLA-4 [134]. Additionally, PD-1 inhibits the transition of the functional CD8^+^ effector into CM T cells [135] by upregulating the pro-apoptotic factor BIM, compromising long-term immune memory potential (Figure 2) [136].

Elevated intra-tumor levels of TIM-3 have also been associated with the suppression of T-cell responses [137] and reduced production of TNF-α, IFN-γ, and GrzB, especially when co-expressed with TIGIT, CTLA-4, or LAG-3 [62,91]. TIM-3 activation by its ligand GAL-9, a member of the β-galactoside-binding protein family [138], induces apoptosis in CD4^+^ and CD8^+^ TILs, dampening T-cell responses and correlating with poor prognosis. Carcinoembryonic antigen cell adhesion molecule 1 (CEACAM1) promotes TIM-3 expression in T cells [139], which paradoxically protects newly activated T cells from premature apoptosis during expansion but contributes to late-stage effector T-cell exhaustion [140]. This can be explained by TIM-3 intracellular localization in late-stage T-cell effectors, enhancing TCR signaling and pro-apoptotic protein expression. In contrast, in newly activated T cells, TIM-3 is surface-expressed and deactivated by CEACAM1 engagement, safeguarding them from RICD.

Metabolites originating from cancer cells and reactive oxygen species (ROS) can also contribute to T-cell apoptosis. For example, kynurenine, a metabolite produced from tryptophan by indoleamine 2,3-dioxygenase in cancer cells, can promote apoptosis in T cells [141]. ROS contribute to activated T-cell death, regulating the immune balance [142]. Indeed, ROS generated during activation enhance FAS expression, reduce anti-apoptotic BCL-2 levels (Figure 2) [143], and hinder NF-κB activation, limiting IFN-γ and TNF-α production [144]. ROS also contribute to Tregs induction and function [145]. While some Treg subsets resist oxidative stress [146], some others undergo apoptosis, generating immunosuppressive adenosine [147].

What emerges is that the dynamic TME milieu involves diverse factors, including immunosuppressive factors and ligands, influencing T-cell apoptosis and compromising anti-tumor responses, yet maintaining immune homeostasis. Understanding these complexities is crucial for identifying effective therapeutic interventions.

### 3.3. Nutrient Competition between Cytotoxic T Cells and Tumour Cells, and Impact on Anti-Tumor T-Cell Functions

Efficient immune responses targeting tumors require precise coordination of T-cell proliferation, apoptosis sensitivity, and metabolic changes to balance robustness and controlled activation. However, our understanding of how cellular metabolism influences T-cell survival remains incomplete. Within the TME, metabolic processes are influenced by substrate availability and environmental signals [148], triggering metabolic reprogramming in distinct lymphocyte subsets [149].

The metabolic dynamics in the TME are significantly influenced by the competition among its main components. The Warburg effect, a key feature of the TME [150], enhances tumor cell survival and proliferation by altering glucose utilization. Research, notably by Voss et al., has shed light on the connection between T-cell metabolism and apoptotic sensitivity in immune responses [85]. Naïve T cells primarily rely on oxidative phosphorylation (OXPHOS) for ATP production, which makes them physiologically inactive until encountering specific antigens [151]. T cells then undergo metabolic reprogramming, shifting to aerobic glycolysis to meet the increased energy demands for proliferation and effector function acquisition [152]. CD28 co-stimulation enhances glucose uptake via Glut1 upregulation (Figure 3), mediated by PI3K/AKT [153,154] and Ras/MEK/ERK [151] signaling pathways, leading to HIF-1α and c-MYC activation. The mTOR pathway, downstream of PI3K/AKT, integrates diverse signals to regulate metabolic reprogramming [153]. IL-2 fosters glycolysis via mTOR, while IL-7 supports Glut1 expression through STAT5-mediated AKT activation, reducing apoptosis and exhaustion [155]. However, heightened glycolytic activity with increased mTORC1 and pentose phosphate pathway (PPP) is associated with reduced self-renewal capacity. Conversely, the activation of the energy sensor AMP-activated protein kinase (AMPK) pathway supports T-cell longevity (Figure 3) [156].

Figure 3 elucidates the main aspects of the TME impacting T-cell metabolism and anti-tumor T-cell mediated responses.

During the peak of an adaptive immune response, T cells undergo an apoptotic RICD process [93], crucial for regulating effector T-cell growth and preventing immunopathology [98]. Accordingly, glycolytic flux correlates directly with RICD sensitivity in human CD8^+^ T cells [157], while a decline in glycolytic activity or exogenous glucose availability significantly reduces effector T-cell sensitivity to RICD. Intriguingly, Larsen’s study revealed that active glycolysis promotes pro-apoptotic FAS-L induction, enhancing RICD in highly glycolytic T cells (Figure 3). However, CD4^+^ T-cell RICD is less influenced by glycolysis, except for the role of glyceraldehyde-3-phosphate dehydrogenase [158], in regulating glycolytic activity and T-cell effector function [159].

Aerobic glycolysis profoundly impacts the TME, primarily through local acidification, inhibiting TIL function [160]. Lactic acid, a glycolytic metabolite, hinders CD4^+^ and CD8^+^ T-cell migration and CD8^+^ T-cell cytolytic function while promoting immunosuppressive Tregs (Figure 3) [161]. Lactic acid also drives the M2-TAM macrophage switch, which promotes T-cell apoptosis via the PD-L1/PD-1 pathway [162]. In addition to glucose, tumor cells in the TME engage in predatory glutamine uptake (Figure 3), which affects anti-tumor responses and viability of TILs, exhibiting increased glutamine metabolism to support proliferation and cytokine production [163].

Highly glycolytic and activated T cells face challenges in maintaining ATP levels in low glucose and glutamine environments due to tumor-dependent deprivation. This triggers AMPK, controlling glutamine-dependent mitochondrial metabolism to sustain T-cell bioenergetics by hindering mTOR activity and T-cell proliferation (Figure 3) [164].

The dynamics of cytokine-removal-induced apoptosis after the primary response significantly influence the size of the memory T-cell subset and is now recognized as a new IC mechanism [5]. To enter the memory pool, T cells must decrease glycolysis to evade RICD and shift to catabolic fatty acid oxidation (FAO) to fuel OXPHOS (Figure 3), favoring T cells with higher spare respiratory capacity (SRC), crucial for immediate recall responses [85]. Accordingly, limiting glycolysis through mTORC1 inhibition enhances memory T-cell development [165,166]. Furthermore, mice with T-cell-specific deletion of TRAF6, a modulator of FAO, display robust CD8^+^ effector T-cell function but show significant deficiencies in generating memory T cells [165]. Notably, PD-1 hampers early glycolytic activity, prompting T cells to shift to FAO to prolong their lifespan, facilitated by increased expression of carnitine palmitoyltransferase 1A (Figure 3) [167].

During the contraction stage following cytokine withdrawal, when withstanding growth factor deficiency [168], T cells upregulate autophagy to fight growth factor deficiency, providing critical ATP production for survival and activation [169]. The heightened sensitivity to CWID in effector T cells from EM compared to CM is attributed to sustained protective autophagy and reduced pro-apoptotic BIM expression [170].

In the glucose-deprived TME, CD8^+^ TILs often rely on FAs as an alternative energy source to sustain their anti-tumor activities. In mouse melanoma models, TILs boost FA breakdown through peroxisome proliferator-activated receptor-α signaling [171]. However, increased FAs uptake via the CD36 transporter impairs CD8^+^ TIL effector functions, heightening ferroptosis and reducing pro-inflammatory cytokine expression [172]. A noteworthy link between FA metabolism and T-cell apoptosis susceptibility has also been identified. Indeed, activated CD4^+^ T cells undergo apoptosis when FA availability is compromised [173]. Conversely, the involvement of FA synthesis in RICD has been suggested by Voss et al., who reported how inhibiting FA synthase (FASN), which synthesizes palmitate from acetyl-CoA and malonyl-CoA, with the C75 inhibitor, reduces CD4^+^ T-cell sensitivity to RICD through a decreased FAS-L expression (Figure 3) [158]. This inhibition is intriguing because FASN inhibition reduces cell growth and PD-L1 expression in leukemia cancer cells [173].

Whether additional impacts can make effector T cells more prone to RICD or redirecting them to an inactive state would protect T cells from RICD during their progression toward a memory phenotype remains uncertain. Arginine, pivotal for T-cell activation and proliferation, is depleted within the TME by MDSC-secreted arginase, hindering T-cell functions. Conversely, elevated arginine levels induce metabolic shifts, favoring OXPHOS and promoting the generation of CM-like cells with enhanced persistence and anti-tumor capabilities, as demonstrated by Geiger et al. (Figure 3) [174].

T_RM_ cells, crucial for tissue defense [71], rely on exogenous FAs for survival and function [175]. Essential FA transporters, such as FABP4 and FABP5, play a critical role in sustaining the longevity and function of CD8^+^ T_RM_ cells, contributing to protective immunity [176]. Notably, in gastric tumors, a reported competition for lipid uptake appears to favor tumor cells, potentially leading to T_RM_ apoptosis (Figure 3).

Disruption of T-cell metabolic requirements leads to impaired expansion and effector functions, pushing them toward exhaustion. Co-inhibitory receptors, including PD-1, CTLA-4, LAG-3, and TIM-3, while inducing T-cell apoptosis, significantly suppress glycolysis and mitochondrial oxidative capacity and promote FAO (Figure 3). Then, persistently exhausted TILs with elevated PD-1 expression may evade RICD by downregulating the TCR signaling and glycolysis [177]. Voss et al. have suggested that the “rationing” of available fuels through catabolic metabolism protects TILs from acute growth factor withdrawal, while inhibition of the TCR signal strength and glycolysis through ICs can safeguard T cells against death from chronic TCR restimulation [85].

## 4. Apoptosis Control to Bypass Immunotherapy Resistance

### 4.1. Unravelling Immunotherapy Challenges: Tumour-Infiltrating T-Cell Apoptosis, Implications for Treatment Resistance and Beneficial Strategies

Recent research has shed light on the involvement of T-cell apoptosis in the resistance to cancer immunotherapies in terms of reduced anti-tumor response, immune evasion by cancer cells, impaired persistence of T cells, and metabolic dysregulation. While ICIs like anti-PD-1 and anti-CTLA-4 mAbs overcome inhibitory signals to unleash T-cell activity against cancer, they also pose a dual role regarding apoptosis regulation [178,179]. While inhibiting apoptosis is vital for sustaining a robust T-cell response, excessive inhibition by ICIs can perpetuate dysfunctional T-cell survival, undermining overall efficacy. For instance, in murine melanoma models, FAS-L-induced T-cell apoptosis has been linked to resistance against CTLA-4 and PD-1 blockade and adoptive cell therapy (ACT). Additionally, overstimulation of CAR-T cells, achieved through CD28 and 4-1BB co-stimulatory domains, can heighten FAS and DR5 expression, increasing CAR-T-cell apoptosis. These findings underscore the importance of finely tuning CAR-T-cell activation to prevent hyperactivation, as excessive glycolytic metabolism enhances T-cell susceptibility to apoptosis [180,181].

While the presence of TILs has traditionally signaled a better prognosis and response to ICB, recent studies unveil a diverse spectrum of T-cell states within tumors, shaped by local inhibitory cues [182]. As stated above, exhaustion is now recognized as a functional alteration rather than outright T-cell inactivation, characterized by a dynamic transition from a “stem-like progenitor” to a “terminally exhausted” state [75], the latter exhibiting a significant but short-lived effector function, often resistant to immunological reinvigoration.

Notably, CD8^+^ T cells re-invigorated by PD-1 blockade have been reported to be characterized by CD28 expression [183]. At variance, TILs displaying terminal differentiation (with high PD-1 and TIM-3) do not benefit from PD-1/PD-L1 blockade [82]. It has been suggested that these cells could possibly be already committed to apoptosis [184], with the engagement of TIM-3 by pro-apoptotic ligands like GAL-9 on tumor cells potentially contributing to RICD [185]. Next-generation ICIs [186], including TIM-3 blockade, offer promise in overcoming this resistance by reducing MDSCs and enhancing T-cell proliferation and cytokine production. Combined targeting of TIM-3 and PD-1 has also shown efficacy in counteracting T-cell exhaustion [187] and overcoming resistance to PD-1/PD-L1 blockade [188]. LAG-3 is expressed by activated T cells, NK, B lymphocytes, and DCs and engages with MHC class II [189], inducing T-cell dysfunction and immune exhaustion while favoring the generation of a Treg phenotype. Notably, LAG-3 blockade can enhance the effect of anti-PD-1 agents and other immunotherapeutic approaches [190].

Recent insights highlight the potent anti-tumor capabilities of T cells bearing memory traits, showing prolonged persistence and heightened anti-tumor activity compared to effector cells, attributed, at least in part, to their inherent resistance to apoptotic RICD, as described above [85,165]. Notably, memory T cells also demonstrate resilience against apoptosis triggered by chemotherapy and radiation, indicating their potential significance in combined therapeutic strategies. This resilience could stem from elevated BCL-2 levels and decreased BIM expression in these cells [191], although further research is required to thoroughly comprehend the mechanisms underlying memory T-cell survival during chemotherapy and radiation.

These observations suggest that fostering the development of tumor-responsive memory T cells in combination with ICB or in the context of ACT and CAR therapies holds promise as a cancer therapy strategy [192]. Indeed, a pre-exhausted phenotype characterized by CD28 expression [79,183] and a high CM/effector T-cell ratio have both been recently reported as predictive of ICB response [192].

Administering cytokines like IL-2, IL-7, IL-15, and IL-21 emerges as a valuable strategy to bolster T-cell survival, proliferation, and overall function within the immunosuppressive TME (Figure 4). Figure 4 indicates factors and mechanisms that can be targeted to mitigate T-cell apoptosis and enhance T-cell survival and the generation of a long-lived memory compartment while bolstering anti-tumor immune responses within the challenging milieu of the TME.

IL-2 is crucial in CD8^+^ T-cell differentiation and Treg homeostasis, yet its continuous stimulation can trigger apoptosis via FAS and FAS-L expression [193]. Although high doses of IL-2 show significant anti-tumor efficacy, careful administration is required due to potential toxicities and the expansion of Tregs. PEGylated IL-2 (NKTR-214) offers a promising alternative, showing improved persistence and superior anti-tumor responses in combination with immunotherapies [193].

IL-7 and IL-15 have been implicated in maintaining CD8^+^ memory T cells (Figure 4) [192]. IL-7 protects T cells from apoptosis, fostering memory generation and IFN-γ production. Recombinant human glycosylated IL-7 (CYT107) has revealed substantial pharmacodynamic changes without significant toxicity, offering a promising avenue for immunotherapy [193].

As reported above, IL-15 promotes T-cell longevity and resistance to TRAIL-associated apoptosis [97] (Figure 4). In contrast to IL-2, IL-15 does not induce the activation of Tregs due to the absence of IL-2Rα chain binding, making it a valuable addition in combination therapies with ACT or ICIs [194].

IL-21 sustains T-cell function upon continuous antigen exposure and acts synergistically with IL-7 or IL-15 to sustain the survival of memory CD8^+^ T cells [195]. Direct inoculation of IL-21 into tumors converts TAMs from an M2 to an M1 phenotype, leading to tumor control (Figure 4) [196]. Despite its brief half-life, early clinical investigations have explored IL-21 potential, often combined with other agents in cancer treatment, like sorafenib, rituximab, or anti-HER-2 agents [197].

The AKT/mTOR pathway plays a crucial role in governing the differentiation of memory T cells [198]. Studies have shown that inhibitors targeting mTOR with molecules, such as rapamycin and tacrolimus, positively influence the generation of memory CD8^+^ T cells (Figure 4), albeit with a potential shift towards a Treg phenotype [199,200]. Additionally, inhibiting AKT has also been found to enhance the expansion of potent tumor-specific T cells with memory properties, indicating the potential of controlling AKT/mTOR signaling as a strategy for promoting memory T-cell development [201].

CAR-T-cell therapies have revolutionized cancer treatment, particularly in CD19^+^ B-cell acute lymphoblastic leukemia [202]. However, challenges persist, including tumor recurrence [203] and premature CAR-T-cell loss [204]. Strategies to address these limitations involve optimizing CAR-T-cell expansion and survival through exposure to IL-7 and IL-15 and enhancing potency [204] through the integration of intracellular signaling domains of co-stimulatory molecules, such as 4-1BB [205,206], and CD27 [121]. These modifications improve CAR-T-cell proliferation, cytokine production, and resistance to apoptosis, enhancing their persistence and anti-tumor activity.

As proposed by Zhu et al., modifying extrinsic apoptosis pathways, like inhibiting the FAS–FAS-L, can prevent TIL apoptosis induced by FAS-L-expressing MDSCs and tumor cells (Figure 4), enhancing CD8^+^ T-cell infiltration, improving T-cell persistence and activity [5], thus synergizing with various T-cell-based immunotherapies [207]. Strategies like deleting FAS or truncating its intracellular death domain hold promise in enhancing CAR-T-cell persistence and anticancer efficacy. Focusing on T-cell subsets or designing CAR-T cells with balanced metabolic reprogramming toward FAO and autophagy, as suggested by Voss et al. [85], could also enhance and prolong anti-tumor responses by modulating T-cell apoptosis sensitivity.

Overexpression of BCL-2 interferes with normal apoptotic signaling, increasing cell survival following IL-2 withdrawal and preserving T-cell capability to target tumor cells (Figure 4) [208]. BCL-2 integration enhances the anti-tumor activity of CAR-T cells in preclinical settings of mouse xenograft lymphoma, improving persistence and reducing RICD [209]. BCL-2 also shows promise in clinical outcomes of CAR-T treatment [210] and resistance to small molecules promoting tumor apoptosis, undergoing intense preclinical and clinical research in hematologic malignancies, mimicking the action of the BH3-only proteins, including venetoclax [211]. Targeting additional anti-apoptotic regulators like BCL-XL (Figure 4) and MCL-1 can influence T-cell differentiation and survival, potentially shaping T-cell-based immunotherapy strategies to amplify immune responses.

Clinical trials employing TRAIL-R2 agonists show promise in reducing MDSC levels. Combining these agonists with other immunotherapeutic approaches holds potential, especially considering CD8^+^ T cells’ insensitivity to TRAIL-induced apoptosis [212].

As reported above, T_RM_ cells rely on exogenous FA metabolism for survival [175], facilitated by FABP4 and FABP5 FA transporters [176]. However, the T_RM_ apoptosis observed in tumors and due to competition for lipid uptake can be mitigated by PD-L1 blockade (Figure 4) [213].

Adenosine plays a significant role in the TME by influencing T-cell apoptosis and metabolism [113,114].

Several preclinical and early clinical studies are exploring antagonists of the A2AR receptor (Figure 4) or agents selectively blocking CD39 and CD73, directly involved in adenosine generation [111], as potential therapeutic agents in different solid tumors, often in combined approaches with ICIs [214], showing promise in creating a more favorable microenvironment.

Hypoxia indirectly induces effector T-cell apoptosis, also promoting Treg activity, a process counteracted by VEGF inhibition [80]. Several receptor kinase inhibitors and mAbs have been developed and are currently employed or undergoing clinical trials [215]. Then, besides inducing tumor vasculature normalization, targeting VEGF can mitigate Treg-mediated apoptosis of intra-tumor effector T cells and has been recognized as a crucial aspect in unraveling the complete potential of ICI blockade, thus holding the potential for improved treatment outcomes [216].

Inhibition of the TGF-β pathway also prevents Treg generation, aiding in mitigating immunoregulatory environments [116]. Several receptor kinase inhibitors, mAbs, and antisense oligonucleotides have been developed to specifically address the TGF-β pathway and are currently undergoing or have undergone clinical trials and are widely discussed by Kim et al. [217]. Metformin, commonly used for diabetes, inhibits T-cell apoptosis by blocking TGF-β-associated Treg differentiation, showing potential in cancer treatment (Figure 4) [218]. As reviewed by Ganjoo et al., according to results obtained in preclinical studies, in a Phase I trial for HNSCC, the combination of metformin with durvalumab has been reported to decrease FOXP3 Tregs while heightening CD8^+^ cell density in the stroma [160]. However, as indicated by Kim et al., the clinical development of the TGF-β pathway antagonists faces significant challenges, mainly focused on minimizing unintended interference with tumor-suppressing functions and inflammatory responses [217].

Intra-tumor ROS contribute to immune balance by inducing apoptosis in activated T cells [142]; thus, scavengers like N-acetyl cysteine (NAC) show promise in enhancing T-cell anti-tumor effects (Figure 4) [219,220].

One avenue of interest lies in exploring epigenetic modifications aimed at promoting memory stemness driving T-cell persistence. Epigenetic regulation is crucial in shaping the generation of memory T-cell subsets. The impact of epigenetic modifiers, such as DNA methylation or HDAC inhibitors, could offer new insights into promoting memory T-cell development and persistence that can improve clinical outcomes following adoptive CAR-T-cell transfer.

Memory T cells arise from a subset of effector cells through de-differentiation and acquisition of specific methylation patterns, while effector-associated genes undergo demethylation. The disruption of the de novo DNA methyltransferase (DNMT3A), which shapes methylation patterns, promotes memory T-cell generation by facilitating demethylation and quicker re-expression of naïve cell-associated genes. Conversely, Tet methycytosine dioxygenase 2, another epigenetic regulator, induces effector cell features by mediating DNA demethylation and the upregulation of effector-associated genes [221]. At the histone level, CD8^+^ T-cell subsets display distinctive patterns of histone marks associated with memory T-cell differentiation. In particular, active histone marks are enriched at memory-related *loci* in naïve and stem CM cells, while effector genes are associated with repressive histone marks. However, a comprehensive delineation of these mechanisms lies beyond the scope of the present review. What emerges is that epigenetic strategies aimed at reprogramming CAR-T cells, encompassing DNMT and HDAC inhibitors, have the potential to promote a memory-like phenotype (Figure 4) and prolonged CAR-T-cell persistence following adoptive transfer, bolstering therapeutic efficacy.

Overall, targeting molecular pathways involved in T-cell apoptosis offers avenues to enhance T-cell endurance and functionality, improving cancer immunotherapy outcomes.

### 4.2. Enhancing Anti-Tumor Response through Metabolic Reprogramming of T Cells

T cells in the TME adapt their metabolism to fulfill the energetic and biosynthetic demands associated with crucial functions like proliferation, cytokine production, and cytotoxicity. Maximizing T-cell longevity and metabolic activity is essential for successful immunotherapy, aiming to reduce apoptosis and enhance functionality through metabolic reprogramming. However, the challenge is achieving molecular specificity in interventions, minimizing impacts on non-target cells for more effective anticancer immunotherapies.

T-cell apoptosis induced by intra-tumor ROS [142] underscores the beneficial effect of glutathione (GSH) in priming T-cell responses and metabolic reprogramming through the activation of mTOR, NFAT, and MYC [222]. Although ROS also support T-cell signaling following antigen recognition [223], a pharmacological boost of mitochondrial metabolic activity related to ROS upregulation has been reported to activate tumor-responsive T cells, amplifying the effectiveness of PD-1 blockade through the activation of AMPK, mTOR, and PGC-1α [224]. Small molecule activators of AMPK, mTOR, or PGC-1α also show a synergistic suppression of tumor growth when combined with PD-1 blockade.

Small molecules Mdivi have been employed in T cells to modify mitochondrial dynamics, leading to enhanced anti-tumor activity by promoting fused mitochondrial structures and inhibiting fission (Figure 5) [225].

Similarly, targeting mTORC1 with rapamycin may improve the mitochondrial energetic profile for better anti-tumor response.

Immunotherapy strategies involving ATC or CAR-T cells also benefit from selecting metabolically robust cells or implementing transgenic modifications to enhance T-cell bioenergetics. Strengthening mitochondrial respiration and FAO can extend the persistence of adoptively transferred T cells, improving tumor control in preclinical models [156].

Co-stimulatory and inhibitory receptors regulate TCR signaling to prevent T-cell energy or excessive activation. Targeting co-stimulatory signals and cytokines also shows promise for inducing metabolic alterations and enhancing overall anti-tumor efficacy in the hostile TME. IC blockade with anti-PD-1 and anti-CTLA-4 mAbs [178,179] improves glucose availability [159], reinvigorating the T-cell anabolic drive, but may paradoxically increase susceptibility to RICD [191]. Recent research highlights the pivotal role of co-stimulatory molecules, including CD28 and CD137, in regulating T-cell metabolic pathways, offering promising targets for immunotherapy to boost T-cell anti-tumor activity. CD28 not only boosts glycolysis but also activates mTOR, enhances mitochondrial performance, and controls mitochondrial cristae tightening in CD8^+^ T cells, affecting respiratory function [226]. Similarly, CD137 enhances glucose metabolism and mitochondrial respiration, contributing to anti-tumor responses by promoting mitochondrial biogenesis dynamics and regulating FAO (Figure 5), thus enhancing anti-apoptotic functions in CD8^+^ T cells [227,228]. Cytokines like IL-2, IL-15, and IL-7 also play pivotal roles in regulating T-cell metabolism. IL-2 drives T cells towards a glycolytic phenotype, while IL-15 OXPHOS and FAO sustain energy production and memory T-cell formation [229]. IL-7 aids T-cell survival and memory T-cell homeostasis by facilitating glycerol uptake, crucial for fueling FAO in memory T cells (Figure 5) [230].

In the TME, the heightened metabolic demands of cancer cells can hinder effector T-cell function by competing for nutrients and producing immunosuppressive metabolites. Understanding how effector T-cell metabolism can be modulated within this context is crucial for developing effective strategies to enhance T-cell persistence. Various molecular intermediates can directly be engaged in the PPP [231]. Inosine, a metabolic substrate, has emerged as a potential alternative fuel source for T cells in glucose-deprived conditions, supporting their growth and function. Studies have shown that supplementation with inosine boosts T-cell-mediated tumor-killing activity in vitro and enhances the efficacy of IC blockade or ACT in mouse models, alleviating metabolic constraints imposed by tumors [232]. An increase in inosine levels can also be achieved through the catabolism of adenosine, facilitated by adenosine deaminase (ADA) (Figure 5), which provides T cells with essential metabolic energy in the form of ATP while reducing the immunosuppressive effects of adenosine [233]. This dual action promotes potent T-cell responses against tumors, potentially improving cancer immunotherapy outcomes [234]. Preclinical studies have demonstrated that ADA overexpression improves CAR-T-cell proliferation, infiltration capacity, control of tumor growth, and overall survival in ovarian carcinoma xenografts and colon cancer solid tumor models [234].

T cells, notably CD4^+^, exhibit heightened RICD dependent on FASN [158]. Inhibiting FASN not only reduces cell growth in various cancer models, including leukemia, but also lowers PD-L1 expression on tumor cells [173], offering a therapeutic avenue to reprogram T-cell metabolism and potentially shield them from tumor-induced immunosuppression. Investigations into metabolic regulators aim to reshape the TME by exploiting differences in metabolic needs among different T-cell subsets, such as Tregs and cytotoxic T cells [235].

As described above, arginine depletion in the TME hampers T-cell activation and proliferation. Exogenous arginine supplementation can shift T-cell metabolism from glycolysis to OXPHOS, countering the Warburg effect [174] and reducing T-cell susceptibility to apoptosis. Indeed, combining the arginase inhibitor CB-1158 with pembrolizumab elevates plasma arginine levels and enhances intra-tumoral CD8^+^ T cells in microsatellite-stable colorectal cancer patients [160].

Besides its described protective effects against T-cell apoptosis, metformin also enhances the migration of adoptively transferred antigen-specific CD8^+^ T cells into tumor sites while preserving T-cell multifunctionality, a mechanism reliant on AMPK activation [236].

As reported above, a “glutamine steal” hypothesis has been proposed, suggesting that the selective blockade of glutamine metabolism in tumor cells could alleviate the metabolic competition for glutamine in the TME. This, in turn, would release glutamine for use by immune cells, potentially improving T-cell survival [163]. Accordingly, DON, a glutamine antagonist or its less toxic derivative DRP-104, could release glutamine for immune cells’ use, enhancing T-cell survival and anti-tumor responses (Figure 5) [163]. DRP-104 is currently under evaluation in clinical trials for safety, pharmacokinetics, and anti-tumor activity [160].

The significant utilization of glucose by cancer cells as an energy source leads to the accumulation of extracellular lactate in the TME that weakens CD8^+^ T-cell cytotoxicity, decreasing IFN-γ production while fostering T-cell apoptosis through increased expression of CTLA-4 and PD-L1 [237]. Potential approaches to modulate the acidic TME, thereby reducing T-cell apoptosis and enhancing immune responses, include, among others, the use of proton pump inhibitors (PPIs), sodium–hydrogen exchanger-1 (NHE1) inhibitors, anti-angiogenic drugs, and agents targeting carbonic anhydrase IX (CA IX) (Figure 5).

These findings underscore the potential of targeting metabolic pathways to control T-cell apoptosis, enhancing anti-tumor activity, and overcoming immunotherapy resistance, offering promising avenues for more effective immunotherapy strategies.

## 5. Interplay between Apoptosis and Metabolism

### 5.1. Metabolites Involved in the Anticancer Response: Escape from Apoptosis through Metabolic Renewal

The core strategy of chemotherapy treatment revolves around inducing apoptosis to combat tumors effectively. To this purpose, anticancer agents such as 5-fluorouracil, etoposide, and staurosporine are commonly utilized. Investigations on metabolic signatures associated with apoptosis have identified alanine and glutamate as pivotal markers for monitoring the response to treatments [238]. Both amino acids are intricately linked to taurine metabolism as they can be synthesized from taurine through processes involving enzymes like pyruvate aminotransferase or taurine2-oxoglutarate transaminase.

Cancer cells adapt by activating various pathways, establishing resistance mechanisms as a result. This renewed cellular metabolism provides an advantageous escape strategy. To illustrate these dynamics, we can consider docetaxel, a drug that blocks cell mitosis by targeting microtubules [239]. Docetaxel can induce apoptosis through BCL-2 phosphorylation, promoting tumor regression [240,241], and is largely employed in combination with many other chemotherapeutic agents to improve therapeutic efficacy. Nevertheless, docetaxel employs several different mechanisms to allow cancer cell survival [242]. Metabolomic studies have provided a chance to monitor its effects, as described by Wang et al. [243]. Among the numerous significant pathways identified in treated cervical cancer cells that are deregulated by docetaxel, ABC transporters have proved the most statistically significant and strongly associated with chemoresistance [244]. From a metabolic perspective, central carbon metabolism is crucially interconnected to glycolysis/gluconeogenesis, cysteine, and methionine metabolism, as well as arginine biosynthesis. These pathways are under the control of RTKs-mediated MAPK signaling or the HER2-mediated PI3K–AKT–mTOR axis [243].

A correlation between cysteine intracellular increase and GSH biosynthesis was investigated in KRAS-mutated lung adenocarcinoma [245]. In the affected patients, the transporter SLC7A11 is highly expressed. This antiporter, which has already been demonstrated to be associated with an increased glutamate secretion in gliomas, exchanges an anionic form of cysteine and glutamate. Hu et al. have reported that targeting this transporter with sulfasalazine or the HG106 inhibitor effectively modulates cystine uptake and intracellular GSH biosynthesis in a mouse model, mainly activating apoptosis through enhancement of oxidative and ER stress. GSH represents a key source for the setting of anticancer strategies. Its synthesis relies on the expression of the two enzymes (ligase and synthetase) which are modulated by the chromatin remodeling factor ARID1A. Its mutation or deletion in cancer cells favors the accumulation of ROS and promotes sensitivity towards apoptosis [246]. The central role of GSH in maintaining the redox status of the cell is managed by glutathione peroxidase 4, which preserves the cellular redox status, counteracting membrane lipid peroxidation. This defense mechanism controls the cellular levels of lipid hydroperoxide derivatives generated by alkoxyl radicals following ferrous iron excess. An imbalance of these mechanisms strongly affects the cells, driving them towards a particular type of apoptosis, ferroptosis [247]. Inducers of ferroptosis have shown promise in eliminating quiescent colorectal cancer cells after chemotherapy with 5-fluorouracil/oxaliplatin, thereby impeding survival and relapse. These drug-resistant cells exhibit a renewed metabolism when exposed to a diet low in carbohydrates and proteins, showing low levels of adenosine and deoxyadenosine monophosphate alongside higher lipidic and organic compounds, in agreement with their lower proliferative capacity [248].

Another cytotoxic agent with significant potential is ascorbic acid, which exhibits antioxidant effects in the extracellular space, enhancing the generation of ROS. When used in combination with ibrutinib, idelalisib, and venetoclax for the treatment of chronic lymphocytic leukemia, ascorbic acid contrasted the pro-survival microenvironmental support provided by bone marrow mesenchymal stem cells, T-cell cues (CD40L + IL-4), cytokines and limited hypoxia conditions [249]. Resistance to ascorbic acid is associated with the catalytic activity of catalase, along with glycolytic enzymes and pyruvate levels.

The M2 isoform of pyruvate kinase (PKM2) is notoriously involved in cancer, catabolizing pyruvate to lactate rather than undergoing complete oxidation via OXPHOS for ATP production. This behavior contributes to the Warburg effect, causing an accumulation of numerous intermediate metabolites during glucose utilization, diverted towards tumor biomass production pathways. In addition, PKM2 is also engaged in mitochondrial functions, as well as the regulation of ROS and apoptosis [250].

PKM2 exists in tetrameric and dimeric isoforms, each with distinct functions. The tetrameric form, with high catalytic activity, directs flux towards OXPHOS, while the dimeric form, characterized by lower enzymatic activity, promotes aerobic glycolysis and possesses transcription factor functions, particularly in association with HIF-1. The dimeric isoform of PKM2 reduces Krebs cycle intermediates by upregulating PDK1, which inhibits PDH, and BCL-2-interacting protein 3 (BNIP3), leading to a decrease in OXPHOS proteins. Conversely, PKM2 promotes the phosphorylation of AMPK, a central regulator of cellular metabolism, which has been described above.

Furthermore, a high PKM2/PKM1 ratio has been reported to promote aerobic glycolysis, and ROS can be accumulated by dysfunctions at the level of the complex I and III of the electron transfer chain. Studies on the effect of ROS have reported contrasting data; however, clearer findings suggest that ROS can impact the PKM2 protein, modifying its degradation and localization. These effects subsist in balance with the influx of metabolites into the PPP pathway to control ROS levels.

Other studies have demonstrated a link between PKM2 and mitochondrial autophagy, mediated by the transcription of *BNIP3* in hypoxic conditions and leading to the release of mitochondrial proteins activating apoptosis. Again, the inhibition of autophagy occurs through the activation of the PI3K–AKT–mTOR axis but can be counteracted by the decrease in AMP/ATP levels through mTORC1 kinase activation. Notably, PKM2 is overexpressed in T cells by the mTOR1-HIF1 signaling, where the dimeric isoform displays function as a transcription factor, leading to the phosphorylation of STAT3 and enhancing Th1 and Th17 differentiation.

The crucial impact of PKM2 on apoptosis is evidenced by its silencing, which regulates the previously described mechanisms functioning as metabolic regulators of cancer cells and promoting cancer cell apoptosis. However, the modulation of mitochondrial functions can counter apoptosis, leading to the onset of resistance mechanisms, such as the suppression of the p53 anti-oncogene under conditions of high oxidative stress or increased expression of the anti-apoptotic factor BCL-XL stabilizing the binding of NF-κB p65 to its promoter.

Due to its relevance, glycolysis is an attractive target for approaches aimed at targeting glucose transport and enzymes to potentiate anticancer therapies.

Another pivotal aspect is the intracellular homeostasis of H^+^ concentration and the involvement of the ion pumps NHE1 and V-ATPase [251]. The NHE1 family controls intracellular pH by transporting H^+^ ions into the extracellular space in exchange with Na^+^ ions, protecting cancer cells from acidification and regulating acid–base homeostasis in balance with bicarbonate transporter and exchanger systems.

Intracellular alkalinization stimulates glycolysis, impacting cell adhesion, tumor development, and migration. Indeed, in breast cancer, NHE1 phosphorylation via AKT increases its affinity for H^+^ ions, promoting H^+^ secretion and creating a microenvironment conducive to proteolytic degradation of the ECM.

Notably, NHE1 displays greater sensitivity compared to other members of the family towards certain inhibitors such as amiloride, benzyolguanidinium-based formatives, cimetidine, clonidine, and harmaline. Conversely, some other inhibitors exhibit a more selective nature compared to NHE1, such as cariporide, compound 9T, and 2-aminophenoxazine-3-one. Overall, this feature renders NHE1 a promising target for anticancer therapies.

Concerning V-ATPase, an ATP-dependent H^+^ pump expressed on both plasma membranes and organelles (e.g., endosomes, lysosomes, vesicles), its expression correlates with endocytosis and is engaged in the activation of proteases (e.g., cathepsins and matrix metalloproteases) during the invasion processes. The pH value determines the assembly of the V-ATPase complex, expressed not only in cancer cells but also in neutrophils, macrophages, and DCs, where it is stimulated by PI3K and mTORC1. Inhibition of V-ATPase can be mediated by intracellular pH increase, which promotes the expression of the pro-apoptotic protein BNIP3 and affects molecule endocytosis, which stimulates migration. However, this may activate lysosomal trafficking and autophagy as a stress response, thus enabling escape from anoikic pathways. E2F1 transcription factor and mTOR play regulatory roles in these processes.

Several V-ATPase inhibitors have been shown to induce apoptosis in cancer cells, including bafilomycin and concanamycin, along with newer compounds such as salicylihalamide A, apicularen A, lobatamide A, oximidine I, cruentaren, NiK12192, PPI SB 242784, and FR202126.

Other proton pumps, such as H^+^/K^+^-ATPase exchange can be involved in gastric cancer. Indeed, studies have shown that the growth of gastric cancer cells is inhibited in xenograft models by pantoprazole [252].

Finally, an acidic pH of the TME can affect the entry of drugs like vinblastine, doxorubicin, vincristine, mitoxantrone, and paclitaxel into cancer cells, reducing their efficacy [253]. Recently, inhibitors targeting CAIX, an enzyme pivotal in pH regulation and frequently overexpressed in various solid tumors, have emerged as a promising frontier in cancer therapy, especially for hypoxic tumors. These inhibitors have demonstrated significant efficacy in preclinical models, notably addressing tumor resistance to conventional cytotoxic agents by mitigating poorly accessible hypoxic regions [254] while potentially improving a T-cell mediated anti-tumor response (Figure 5).

We summarized the relationship between drugs and metabolism in the context of apoptosis or its escape in Table 1.

### 5.2. Extracellular Metabolites and Their Relation with the TME

Following the extensive utilization of aerobic glycolysis by tumor cells, the accumulation of lactic acid in the TME influences cellular components and their interactions [255]. Due to extreme variability in the content and typology of non-tumor cells in the TME, including adipocytes, fibroblasts, the tumor vascular system, lymphocytes, DCs, and tumor-related-CAFs, several synergistic or contrasting interrelationships emerge, shaping unique niches conducive to tumor development.

The impact on the genesis of new blood vessels for vascularization is critical, supporting tumor growth and metastasis. However, the influence of glycolysis extends beyond this, affecting tumor immunity and outlining an immunosuppressive profile within the TME. On the one hand, while glycolysis can increase the expression of PD-L1, a prerequisite for a good immunotherapeutic response, on the other hand, it also fosters a favorable inflammatory milieu for certain tumors, such as breast cancer, characterized by an enrichment in Th2 cells and macrophages and a reduction in cytotoxic immune cells.

Lactate contributes to the acidification of the TME, promoting the transition of macrophages from the tumor-suppressive M1 phenotype to the oncogenic M2 phenotype. In addition, cancer cell-produced fibronectin 1 induces metabolic changes in macrophages, activating glycolysis and the enzyme PKM2.

Acting as a transcription factor, PKM2 promotes the expression of several ILs and proteins (IL-1β, IL-12p70, TNF-α, HLA-DR, and PD-L1), thereby polarizing macrophages via HIF-1α, decreasing their anticancer action. Aberrant intracellular glucose metabolism helps cancer cells to survive by controlling apoptosis. This condition, strictly associated with hypoxia, sees the implication of TAMs in the accumulation of endothelin, vascular endothelial growth factors, and ILs, supporting vascularization and metastasis. High levels of lactate in the TME limit the response of cytotoxic T lymphocytes and NK cells. The acidic nature of the TME affects lymphocyte functions, such as activation and migration, facilitating cancer cell evasion from immunosurveillance. DCs are also affected, with TME acidity suppressing their activation and antigen expression. Increased PD-1 expression in Tregs modulates immune responses and promotes metastasis.

The immune response is also driven by the levels of ATP released into the TME during chemotherapy-induced cell death [256]. This release attracts macrophages and DCs, which are activated by G protein-coupled P2Y receptors, stimulating chemotaxis, phagocytosis of dying cells, and sustained inflammation fostered by antigen processing and presentation to T cells into lymph nodes. In the same cells, ATP also stimulates P2X7 receptors, leading to the formation of the inflammasome complex and secretion of IL-1β, which polarizes T cells and enhances their cytotoxic capabilities. These processes can lead to anticancer effects or contribute to hyper-inflammatory conditions, favoring vascularization and dissemination of metastases.

ATP and its metabolites, such as ADP or AMP, are hydrolyzed mainly in the extracellular space by several ectonucleotides, generating adenosine, which acts on lymphocytes, particularly T cells, via A2A/A2BR, as stated above. Extracellular ATP concentration can hinder the efficacy of chemotherapy. Moreover, ATP, ADP, and AMP can be released by cancer cells through various mechanisms, including endocytosis and plasma membrane channels, particularly under conditions of drug-induced apoptosis (e.g., with etoposide or doxorubicin). Indeed, the amount of intracellular ATP is depleted more rapidly by apoptotic cells due to mitochondrial damage, so its influx increases through permeable channels. Conversely, certain extracellular metabolites, such as AMP, GMP, and oxidized glutathione, can induce apoptosis, representing a significant pathway for targeting cancer cells [257].

The TME not only serves as a repository for directly secreted metabolites but also contains extracellular vesicles (EVs), which facilitate inter-cellular communication and can originate either locally or from distant sites. In the TME, EVs can induce EMT, promote invasive features, or facilitate the formation of metastatic niches, thereby aiding in the colonization of circulating tumor cells [258]. An interesting impact on metabolism is exploited by miRNAscontaining EVs produced from cancer cells, which interfere with neighboring cells, leading to reduced glucose consumption and increased availability for tumor growth. For instance, miR-122 contained in EVs secreted by tumors can influence the glycolytic pathway in fibroblasts by affecting pyruvate kinase and their glycolytic pathway. In addition, cancer-derived EVs containing miR-210 stimulate angiogenesis and can transport various metabolites, including Krebs cycle intermediates, proteins, and amino acids to support tumor energy requirements.

In the context of the TME, adipose cells have a relevant role [259], not only through the secretion of inflammatory cytokines, adipokines (such as leptin), and hormones or proteases affecting the extracellular matrix but also by releasing molecules, such as miRNAs and metabolites, associated with the loss of the TME anticancer properties. Among them, some miRNAs target transcripts encoding proteins involved in extracellular matrix maintenance (e.g., collagenases, integrins, laminins) or the TGF-β signaling pathway, with an oncogenic or anti-oncogenic function, respectively. Notably, mimicking breast cancer, co-culture studies using myoepithelial cells in the presence of multipotent adipose or stem cells have demonstrated the impact of adipose cells on the TME [260].

## 6. Conclusions and Future Perspective for Cancer Therapy

In conclusion, unraveling the complex interplay between apoptosis, TME dynamics, and anti-tumor T-cell responses is pivotal for advancing cancer therapy. Despite the formidable challenges faced in inducing apoptosis in cancer cells due to resistance mechanisms, the exploration of multifaceted approaches becomes imperative. Strategies such as targeting key apoptosis regulatory molecules, promoting ICD, and modulating immunosuppressive cells and metabolic pathways hold promise in overcoming resistance and enhancing therapeutic outcomes.

Nevertheless, the metabolic competition within the TME significantly influences anti-tumor T-cell functions and therapeutic responses, emphasizing the need to understand metabolic dynamics and nutrient competition for devising effective immunotherapeutic strategies.

Integration of apoptosis induction, immunogenicity enhancement, and TME modulation in a multifaceted approach offers a promising path forward in cancer therapy. However, as we delve deeper, the delicate balance between apoptosis induction and T-cell survival emerges as a critical consideration. Indeed, while leveraging apoptosis in cancer cells shows potential, preserving T-cell functionality within the TME is imperative to selectively eliminate cancer cells while safeguarding T-cell survival and function for optimizing immunotherapy outcomes. This necessitates selective targeting of apoptotic pathways in cancer cells while sparing normal tissues to minimize off-target effects and toxicity. Moreover, strategies to enhance T-cell survival must be tailored to promote anti-tumor immunity without exacerbating autoimmune responses or inducing T-cell exhaustion.

To this purpose, patient stratification based on tumor biology, immune status, and genetic profiles can optimize the selection of therapeutic interventions. Personalized medicine approaches can identify patients most likely to benefit from apoptosis-inducing therapies and those who may require additional strategies to enhance T-cell function. Biomarkers indicative of apoptosis sensitivity or immune responsiveness can guide treatment decisions, leading to improved clinical outcomes. Overall, our review provides a cutting-edge examination of harnessing apoptosis for cancer treatment. The novelty lies in its in-depth analysis of the multifaceted aspects of apoptosis within the TME, underscoring the necessity of a comprehensive approach aimed at targeting cancer cells while preserving a durable T-cell-mediated anti-tumor response. By delving into the critical role of apoptosis and its intricate interplay with diverse immunomodulatory pathways, we unveil novel insights that promise to advance therapeutic outcomes and overcome the challenges posed by cancer progression and resistance.

## Figures and Tables

**Figure 1 cells-13-00924-f001:**
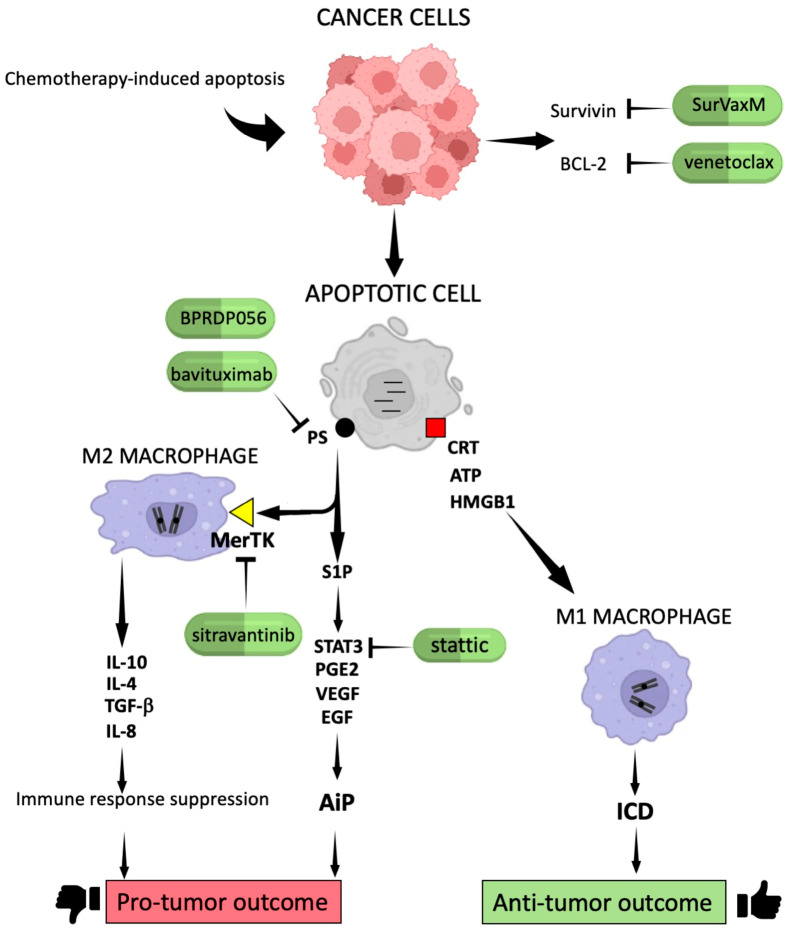
The fate of cancer cells after chemotherapy. Treatment with anticancer drugs stimulates apoptosis in cancer cells that in turn release a pool of molecules in the TME. They may promote tumor cell survival and inhibition of immune responses or activate M1 macrophages and ICD, leading to cancer regression. Pointed arrows mean activation, while flat-tipped arrows mean inhibition. S1P: sphingosine-1 phosphate; CRT: calreticulin; HMGB1: high-mobility group box 1; AiP: apoptosis-induced proliferation; ICD: immunogenic cell death. This figure was created on Biorender.com (accessed on 11 April 2024).

**Figure 2 cells-13-00924-f002:**
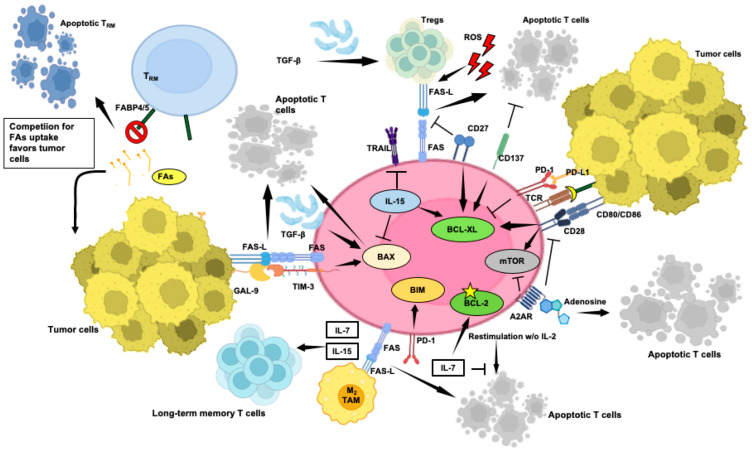
Selected aspects of the intricate network occurring within the TME that contribute to or prevent T-cell apoptosis. Key factors, such as inhibitory ligands and immune suppressive cells, along with anti-apoptotic co-stimulatory signals, are depicted, highlighting their roles in controlling anti-tumor immune response. Understanding these mechanisms is critical for developing targeted therapies to enhance T-cell survival and improve immunotherapy outcomes in cancer patients. Pointed arrows mean activation, while flat-tipped arrows mean inhibition. FAs: fatty acids. This figure was created on Biorender.com (accessed on 3 May 2024).

**Figure 3 cells-13-00924-f003:**
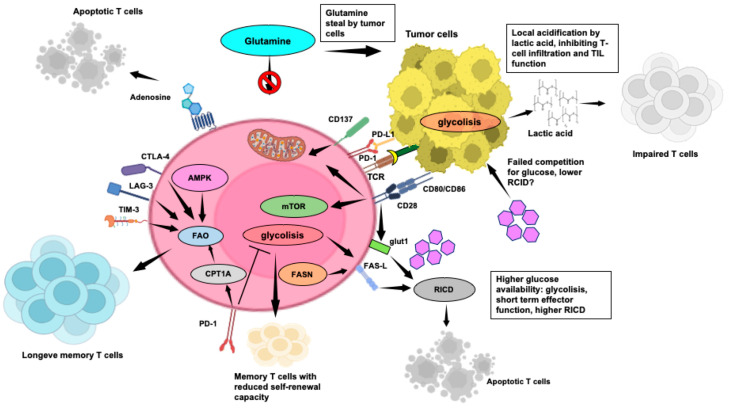
Key aspects of the TME impacting T-cell metabolism and anti-tumor T-cell-mediated responses, including metabolic competition, nutrient availability, immunosuppressive factors, and signaling pathways that shape the metabolic and functional state of T cells. Understanding these aspects is crucial for devising strategies to enhance T-cell function and improve immunotherapy efficacy in cancer treatment. Pointed arrows mean activation, while flat-tipped arrows mean inhibition. This figure was created on Biorender.com (accessed on 3 May 2024).

**Figure 4 cells-13-00924-f004:**
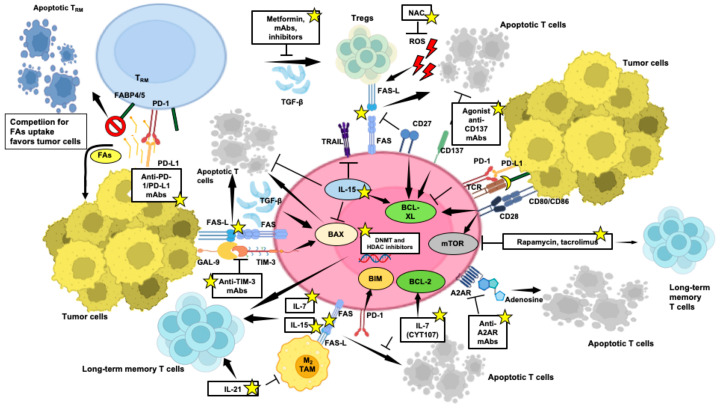
Targeted strategies to counter T-cell apoptosis within the TME, crucial for enhancing anti-tumor immune responses. Potential therapeutic interventions aimed at bolstering T-cell survival and memory potential and efficacy in tumor eradication are indicated by yellow stars. Pointed arrows mean activation, while flat-tipped arrows mean inhibition. This figure was created on Biorender.com (accessed on 3 May 2024).

**Figure 5 cells-13-00924-f005:**
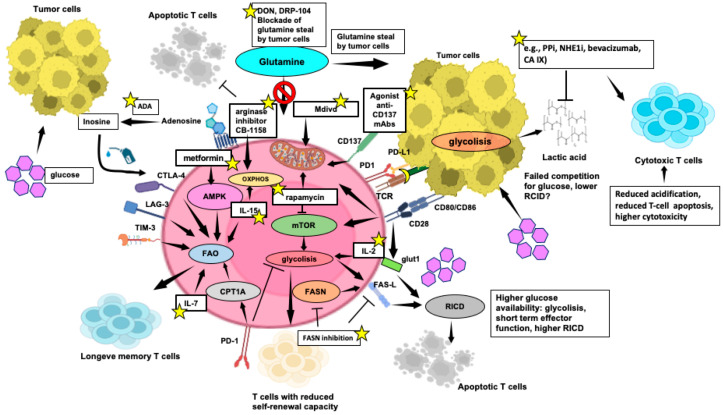
Selected strategies of metabolic reprogramming enhancing T-cell function within the TME. Key mechanisms and metabolic pathways that can be modulated to enhance the metabolic fitness and effector functions of T cells, crucial for effective tumor elimination, are indicated by yellow stars. Pointed arrows mean activation, while flat-tipped arrows mean inhibition. This figure was created on Biorender.com (accessed on 3 May 2024).

**Table 1 cells-13-00924-t001:** Relationship among different compounds and metabolites in distinctive cancer models.

Drug/ Compound	Metabolite/ Enzyme/Pathway	Cancer/Model	Reference
staurosporine 5-fluorouracil etoposide	increase of alanine, arginine, glutamate, and acetyl carnitine (taurine metabolism)	HEK293 and HepG2 cells	[238]
docetaxel	glycolysis/gluconeogenesis, cysteine, and methionine metabolism, and arginine biosynthesis	cervical cancer	[243]
sulfasalazine HG106 inhibitor	increase in intracellular cysteine and glutamate secretion (GSH biosynthesis)	patients affected by KRAS-mutated lung adenocarcinoma	[245]
fasting combined with 5-fluorouracil oxaliplatin	low levels of adenosine and deoxyadenosine monophosphate, high levels of lipidic and organic compounds	colorectal cancer cells	[248]
ascorbic acid combined with ibrutinib, idelalisib, and venetoclax	transport of H^+^ ions and generation of ROS	chronic lymphocytic leukemia	[249]
inhibitors of NHE: amiloride, benzyolguanidinium, cimetidine, clonidine, harmaline cariporide, compound 9T, 2-aminophenoxazine inhibitor of NHE-3-one, ethylisopropylamiloride, hexamethylamiloride, dimethylamiloride	HCO_3_^−^ and H^+^-based transporting systems	human cancer cells	[251]
V-ATPase inhibitors: bafilomycin, concanamycin, salicylihalamide A, apicularen A, lobatamide A, oximidine I, cruentaren, NiK12192, PPI SB 242784, and FR202126	H^+^ gradient produced by H^+^-ATPase	breast cancer, esophageal carcinoma, lung carcinoma, hepatocellular and pancreatic carcinoma, oral squamous cell carcinoma, sarcoma and other solid tumors	[251]
pantoprazole	H^+^/K^+^-ATPase	xenograft model of nude mice with gastric cancer	[252]
vinblastine, doxorubicin, vincristine, mitoxantrone, paclitaxel in combination with CA IX inhibitors	H^+^ in the extracellular space	many cancer cell lines; xenograft model; patients	[254]

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
