# Peer review of "Apoptosis, a Metabolic “Head-to-Head” between Tumor and T Cells: Implications for Immunotherapy"

_cells, 2024, doi:10.3390/cells13110924_

Round 1

Reviewer 1 Report

Comments and Suggestions for Authors

1. I would like you to add comments on the subchapters reviewed in the manuscript and their relationship to cancer stem cells.

2. I suggest a list of abbreviations and their meanings.

Author Response

Reply to the Reviewer 1

Reviewer’s comments

  1. I would like you to add comments on the subchapters reviewed in the manuscript and their relationship to cancer stem cells.

Answer: As suggested, we included in the text comments concerning cancer stem cells in the paragraph 2.1. Molecular signals generated by apoptotic cells in cancer, line 412.

  1. I suggest a list of abbreviations and their meanings.

Answer: To comply with the reviewer's requests, since the journal does not provide a list of abbreviations in the format-template, we added it as Supplementary file 1, including all abbreviations reported.

Reviewer 2 Report

Comments and Suggestions for Authors

The article provides a comprehensive view of using apoptosis to treat cancer, emphasizing the importance of a multifaceted approach to overcome the challenges in cancer therapy. The authors have discussed about various strategies to selectively induce apoptosis in cancer cells without harming other critical aspects of immune response, particularly the T-cells, which are important for a strong and lasting defense against tumors. Major points that the authors need to address are as follows:

1. The novelty of the article should be clearly highlighted as few reviews have already been published on this topic.

2. More important references from last few years should be discussed to improve visibility and quality of current work.

3. The search strategy used for the literature review should be indicated.

4. The authors should discuss about the important genetic and epigenetic mechanisms that can regulate apoptosis as well as immune evasion in cancer cells and aid in identification of novel therapeutic targets.

5. The authors should also highlight few strategies to not only kill tumor cells but also ensure long-lasting immunity. This could involve discussion about how to promote memory T-cell formation and function within the TME.

6. The manuscript should be carefully checked for typographical errors.

Comments on the Quality of English Language

Minor editing needed.

Author Response

Reply to the Reviewer 2

Reviewer’s comments

The article provides a comprehensive view of using apoptosis to treat cancer, emphasizing the importance of a multifaceted approach to overcome the challenges in cancer therapy. The authors have discussed about various strategies to selectively induce apoptosis in cancer cells without harming other critical aspects of immune response, particularly the T-cells, which are important for a strong and lasting defense against tumors. Major points that the authors need to address are as follows:

  1. The novelty of the article should be clearly highlighted as few reviews have already been published on this topic.

Answer: The novelty of our review has been highlighted in the final considerations (1.Introduction, line 97; 6.Conclusion and future perspective for cancer therapy see lines 1311), pointing out how it provides an in-depth analysis of the multifaceted aspects of apoptosis within the TME, underscoring the necessity of a comprehensive approach aimed at targeting cancer cells while preserving a long-lasting T-cell-mediated antitumor response.

  1. More important references from last few years should be discussed to improve visibility and quality of current work.

Answer: In response to the reviewer's feedback, we've updated the references in our revised manuscript to include more recent sources. However, recognizing the relevance of certain older references, we have chosen to retain them when appropriate. Please, note that in the marked-up version we have left only the most significant changes, to facilitate reading and avoid possible errors in the new format, so we “accepted” the new and modified references in the text.

  1. The search strategy used for the literature review should be indicated.

Answer: With this aim, we consulted Pubmed‐NCBI database (https://pubmed.ncbi.nlm.nih.gov/) formulating the following query: With this aim, we consulted Pubmed‐NCBI database (https://pubmed.ncbi.nlm.nih.gov/), formulating the following query: “((“apoptosis”) OR (“apoptotic”)) AND ((“cancer”) OR (“tumor”)) AND ((“tumor microenvironment” OR /“extracellular matrix”)) AND ((“me-tabolism”) OR (“metabolome”) OR (“metabolic”)) AND (“immune cell system”)[MeSH Terms]))”. Further research included ((T-cell apoptosis)) AND (“anti-tumor response”) AND (“T-cell homeostasis”) OR (“T-cell metabolism”). The investigation focused on sev-eral key aspects: fostering anti-tumor T-cell mediated responses, the influence of T-cell metabolism on susceptibility to apoptosis, and strategies for enhancing response to im-munotherapies. This involved safeguarding T cells from apoptosis while bolstering the generation of long-lived memory T cells, optimizing the efficacy of IC inhibitors (ICIs) and Chimeric Antigens Receptor Cells-T (CAR T) cell therapy. Throughout the manuscript, we focused on the latest research findings, emphasizing their potential therapeutic applications. We consistently cited relevant literature to support our discussions, ensuring that our analysis aligns with current advancements in the field.

  1. The authors should discuss about the important genetic and epigenetic mechanisms that can regulate apoptosis as well as immune evasion in cancer cells and aid in identification of novel therapeutic targets.

Answer: The functions of genetic and epigenetic mechanisms, as well as some transcription actors were already described in the original manuscript, for example in the paragraph 5.1 in relation to with apoptosis and metabolism. However, we have improved this topic, including an extensive discussion starting from the line 234. In addition, we included a paragraph, from line 950, concerning the epigenetic regulation in the generation of memory T cells.

  1. The authors should also highlight few strategies to not only kill tumor cells but also ensure long-lasting immunity. This could involve discussion about how to promote memory T-cell formation and function within the TME.

Answer: In the original manuscript, we extensively discussed the pivotal role of generating a long-lived memory T-cell subset and its intricate link with apoptosis sensitivity, highlighting the influence of key cytokines such as IL-7, IL-15, and IL-21, as well as metabolic enzymes (refer to chapters 3.2, 3.3, 4.1, etc). However, we acknowledge the Reviewer about the necessity for a more thorough examination of this aspect. Therefore, in the revised version, we have placed greater emphasis on the importance of fostering the development of enduring memory T-cells, elucidating specific strategies for achieving this, and underscoring their significance in mitigating T-cell apoptosis and bolstering a sustained antitumor response. Please find new sentences at line 832, 880, 950.

  1. The manuscript should be carefully checked for typographical errors.

Answer: We have corrected typographical errors.

Reviewer 3 Report

Comments and Suggestions for Authors

In the present manuscript, the authors comprehensively discuss apoptosis, its importance in cancer, and its implications for immunotherapy. It covers various critical topics on the active role of apoptosis in inducing anti-tumor immunity, how cancer cells escape apoptosis, the impact of apoptosis in maintaining anti-tumor T-cell responses, various challenges in immunotherapy and possible strategies to overcome the challenge, and how to enhance anti-tumor T cell responses through metabolic programming. Cancer therapy has been one of the challenging research avenues. Cancer biology needs to be understood properly to discover a very efficient and effective therapeutic intervention. Understanding apoptosis and its crosstalk with various immunomodulatory pathways to successfully remove or eliminate cancer cells is very essential in the current cancer research. Hence, the information provided in the manuscript is commendable. Overall, the manuscript is drafted very well by gathering all related information. However, I believe the manuscript would have been better if the authors organized or included some simplified figures for easy understanding. I feel the inclusion of extensive information without much graphical representation or figures compromises the perspicuousness of the review. Understanding various mechanisms, crosstalk, factors, proteins, receptors, ligands, metabolites, cytokines, immune cells, and many more without clear and simple graphical representations will be very difficult for readers.

Suggestions:

1. Figure 1 represents some of the general pathways that determine the fate of the cancer cells. The comprehensive details of the pathways are highlighted in the section "Crostalk between apoptotic and immune cells affects the fate of cancer: line 87-179". If the authors mention the figure in this section in every related area will make it easier to understand the literature.

2. The statement "but chemotherapy-induced apoptosis cannot always trigger an immune response: line 175-176" is not very clear considering the information provided above. All the chemotherapy mentioned above seemed to have a positive impact on the immune response. Either the authors can add some examples of chemotherapy that promotes therapy resistance, or the statement can be moved to section 2.2 where the negative impacts of the chemotherapy are mentioned (as in Lines 209-221). 

3. Lines 249-252 and 262-265 can be put together to avoid redundancy.

4. Many mechanisms highlighted under the section 3.2 are depicted in the Figure 2. It would be better if the figure is mentioned in all the related sections so the reader can refer to the figure at the same time. Otherwise, understanding such complex mechanisms will be very difficult. 

5. Figure 2 looks very complex. It includes all the pathways being discussed in the review, and the figure is also mentioned nowhere even in the related literature. The figure just pops up at the end showing every pathway in a very complex form. Reorganizing based on the related section will make it more catchy and much easier to understand. 

6. The review provides several important drugs and metabolites, and their impact on cancer therapy. It would be good if a well-organized table is provided including those drugs/metabolites/compounds, mode of action, and the type of cancer involved.

7. RCID in line 579 should be RICD.

Author Response

Reply to the Reviewer 3

Reviewer’s comments

In the present manuscript, the authors comprehensively discuss apoptosis, its importance in cancer, and its implications for immunotherapy. It covers various critical topics on the active role of apoptosis in inducing anti-tumor immunity, how cancer cells escape apoptosis, the impact of apoptosis in maintaining anti-tumor T-cell responses, various challenges in immunotherapy and possible strategies to overcome the challenge, and how to enhance anti-tumor T cell responses through metabolic programming. Cancer therapy has been one of the challenging research avenues. Cancer biology needs to be understood properly to discover a very efficient and effective therapeutic intervention. Understanding apoptosis and its crosstalk with various immunomodulatory pathways to successfully remove or eliminate cancer cells is very essential in the current cancer research. Hence, the information provided in the manuscript is commendable. Overall, the manuscript is drafted very well by gathering all related information. However, I believe the manuscript would have been better if the authors organized or included some simplified figures for easy understanding. I feel the inclusion of extensive information without much graphical representation or figures compromises the perspicuousness of the review. Understanding various mechanisms, crosstalk, factors, proteins, receptors, ligands, metabolites, cytokines, immune cells, and many more without clear and simple graphical representations will be very difficult for readers.

Suggestions:

  1. Figure 1 represents some of the general pathways that determine the fate of the cancer cells. The comprehensive details of the pathways are highlighted in the section "Crostalk between apoptotic and immune cells affects the fate of cancer: line 87-179". If the authors mention the figure in this section in every related area will make it easier to understand the literature.

Answer: We have repositioned Figure 1 at the end of the section as suggested by the reviewer and added the following sentence: “All these concepts have been schematized in Figure 1 for a better understanding, in which we have also included the drugs useful to affect them, as discussed below.” Furthermore, we have corrected an error in the figure and mentioned the Figure 1 more times in the section.

  1. The statement "but chemotherapy-induced apoptosis cannot always trigger an immune response: line 175-176" is not very clear considering the information provided above. All the chemotherapy mentioned above seemed to have a positive impact on the immune response. Either the authors can add some examples of chemotherapy that promotes therapy resistance, or the statement can be moved to section 2.2 where the negative impacts of the chemotherapy are mentioned (as in Lines 209-221). 

Answer: As suggested, the paragraph was modified as follows: “In fact, the treatment with tamoxifen, paclitaxel and other drugs may promote therapy resistance by M2-like macrophage activation in different cancer types. Therefore, strategies addressed to target M2-like macrophages are crucial to overcome antitumor drug resistance [Wang S et al, 2024]”.

  1. Lines 249-252 and 262-265 can be put together to avoid redundancy.

Answer: To avoid redundancy, we have rephrased it as follows: “The inhibition of MerTK induces the accumulation of cancer apoptotic cells and triggers a type I interferon response” (line 384).

  1. Many mechanisms highlighted under the section 3.2 are depicted in the Figure 2. It would be better if the figure is mentioned in all the related sections so the reader can refer to the figure at the same time. Otherwise, understanding such complex mechanisms will be very difficult. 

Answer: Acknowledging the reviewer's astute observation, we have addressed the complexity of Figure 2 by dividing it into four separate figures. This reorganization aligns each figure with the specific mechanisms discussed in corresponding paragraphs, enhancing clarity, and facilitating better comprehension for readers.

  1. Figure 2 looks very complex. It includes all the pathways being discussed in the review, and the figure is also mentioned nowhere even in the related literature. The figure just pops up at the end showing every pathway in a very complex form. Reorganizing based on the related section will make it more catchy and much easier to understand. 

Answer: Following the reviewer's suggestion, we've restructured Figure 2 and reported in the text the newly arranged figures in alignment with observations from relevant literature. We trust that these revisions provide a clearer description and address the reviewer's concerns effectively.

  1. The review provides several important drugs and metabolites, and their impact on cancer therapy. It would be good if a well-organized table is provided including those drugs/metabolites/compounds, mode of action, and the type of cancer involved.

Answer: This suggestion helps us to summarize the concept that some drugs could be useful to control metabolic changes in the perspective of anticancer therapies. Also, we included the Table 1 at the end of the paragraph 5.1.

  1. RCID in line 579 should be RICD.

Answer: We have corrected it.

Round 2

Reviewer 2 Report

Comments and Suggestions for Authors

The authors have addressed all my concerns.

Comments on the Quality of English Language

Minor editing needed.